ecology/evolution/palaeontology

microwear, reptile, tooth row, out-lever, mechanical advantage, aspect ratio

**Author for correspondence:**
Jordan Bestwick
e-mail: j.bestwick@bham.ac.uk

# Dental microwear texture analysis along reptile tooth rows: complex variation with non-dietary variables

Jordan Bestwick[1,2], David M. Unwin[3], Donald M. Henderson[4] and Mark A. Purnell[2]

[1]School of Geography, Earth and Environmental Sciences, University of Birmingham, Edgbaston, Birmingham B15 2TT, UK
[2]Centre for Palaeobiology Research, School of Geography, Geology and the Environment, University of Leicester, Leicester LE1 7RH, UK
[3]Centre for Palaeobiology Research, School of Museum Studies, University of Leicester, Leicester LE1 7RF, UK
[4]Royal Tyrrell Museum of Palaeontology, Drumheller, Alberta, Canada T0J 0Y0

 JB, 0000-0002-1098-6286; DMU, 0000-0002-9312-2642; MAP, 0000-0002-1777-9220

Dental microwear texture analysis (DMTA) is a powerful technique for reconstructing the diets of extant and extinct taxa. Few studies have investigated intraspecific microwear differences along with tooth rows and the influence of endogenous non-dietary variables on texture characteristics. Sampling teeth that are minimally affected by non-dietary variables is vital for robust dietary reconstructions, especially for taxa with non-occlusal (non-chewing) dentitions as no standardized sampling strategies currently exist. Here, we apply DMTA to 13 species of extant reptile (crocodilians and monitor lizards) to investigate intraspecific microwear differences along with tooth rows and to explore the influence of three non-dietary variables on exhibited differences: (i) tooth position, (ii) mechanical advantage, and (iii) tooth aspect ratio. Five species exhibited intraspecific microwear differences. In several crocodilians, the distally positioned teeth exhibited the 'roughest' textures, and texture characteristics correlated with all non-dietary variables. By contrast, the mesial teeth of the roughneck monitor (*Varanus rudicollis*) exhibited the 'roughest' textures, and texture characteristics did not correlate with aspect ratio. These results are somewhat consistent with how reptiles preferentially use their teeth during feeding. We argue that DMTA has the potential to track mechanical and behavioural differences in tooth use which should be taken into consideration in future dietary reconstructions.

# 1. Introduction

Three-dimensional dental microwear textural analysis (DMTA) is a powerful technique for quantitatively reconstructing the diets of extant and extinct taxa. DMTA quantifies the sub-micrometre scale tooth surface textures that form during food consumption using standardized textural parameters (scale-sensitive fractal analysis [1,2] and/or ISO 25178-2 areal texture parameters [3]), to identify dietary differences between populations and/or species [4–7]. DMTA has most often been applied to terrestrial mammals (see [8] for a review), but has also been applied to fishes [5,9] and more recently, to reptiles [10,11]. Furthermore, DMTA has been shown to be effective at discriminating diets between species or populations, even with low sample sizes [12–14]. DMTA therefore provides a valuable tool for providing robust insight into the levels of niche partitioning and resource competition for a huge range of modern taxa [6,9,14–18], and for reconstructing extinct food webs from the diets of fossil taxa [19–28].

Fewer studies, however, have explicitly investigated intraspecific tooth microwear texture differences within tooth rows and the influence of endogenous non-dietary variables on microwear textures [29–36]. Non-dietary variables include: (i) tooth position, (ii) bite force, and (iii) tooth shape. DMTA of terrestrial mammals samples the homologous occlusal facets of the pre-molars or molars (see [37] for a review), which provides a standardized sampling strategy for these taxa. However, taxa that possess non-occlusal dentitions, including reptiles, fishes and aquatic mammals such as odontocete whales, have very few homologous tooth landmarks or wear facets to sample from [34,38–40]. No standardized sampling locations therefore exist for DMTA of non-occlusal dentitions [34]. A recent analysis of lepidosaur reptile tooth rows found that the anterior-most and posterior-most tooth textures were significantly 'rougher' than middle teeth textures in several species [11]. While these results were interpreted as evidence that standardized sampling locations were required for DMTA of reptiles, only two texture parameters were examined in the analyses, and there was no consideration for other non-dietary variables besides tooth position [11]. Investigating both intraspecific microwear texture differences along tooth rows and the influence of endogenous non-dietary variables could have significant implications for dietary reconstructions of extant and extinct taxa with non-occlusal dentitions. First, it may further indicate if standardized sampling positions are actually needed [30,34]. This would be especially beneficial for fossil taxa that are primarily known from isolated teeth. Second, it may identify texture parameters that are more or less influenced by non-dietary variables and are thus less or more useful for dietary discrimination, respectively [34]. Third, microwear differences that are only influenced by dietary variables may indicate that taxa use different parts of their jaws to consume different foodstuffs [31], which may provide better characterizations of the ecological roles that taxa perform within ecosystems [32,41].

Across all major vertebrate groups, bite force increases along a mesial–distal axis as bite points are located closer to the jaw adductor musculature [42–46]. Although mechanisms of microwear formation are not fully understood [6,47,48], bite force differences along tooth rows during feeding may be expected to have some influence on intraspecific microwear texture differences. Theoretical models that calculate actual bite force are cost-effective but require detailed knowledge of the cross-sectional area and insertion angles of the adductor musculature [44], which can be difficult to obtain from extant skeletal specimens and fossil specimens [49,50]. However, many taxa with non-occlusal dentitions, such as reptiles, do not chew foodstuffs but rather swallow items whole or rudimentarily crush or tear items into bite-size pieces [51,52]. The opening and closing of reptile jaws are therefore often viewed as simple levers where bite force is calculated according to the positions of sampled teeth relative to the fulcrum (the position at which levers turn), known as a mechanical advantage [42,49,50]. Mechanical advantage is therefore a straightforward, yet informative, proxy for understanding mechanical forces that occur along tooth rows during feeding [42,49,50,53].

With respect to tooth shape, DMTA is not underpinned by assumptions of close relationships between the morphology and inferred functions of teeth [5,9,54]. However, few studies have explicitly taken tooth shape into consideration when investigating intraspecific microwear differences [36,55], and none have done so for taxa with non-occlusal dentitions. As a taxonomic grade, reptiles exhibit varying levels of tooth heterodonty. Some species exhibit what are regarded as homodont dentitions (e.g. *Gavialis gangeticus*, the gharial, and *Varanus salvator*, the Asian water monitor lizard), while other species possess mesial teeth that are markedly taller and slenderer than their shorter and rounder distal teeth (e.g. *Alligator mississippiensis*, the American alligator, and *Varanus niloticus*, the Nile monitor lizard) [39,40,56]. From a mechanical viewpoint, tooth shape determines the size of tooth–food contact areas and, consequently, the pressures that teeth are subjected to during feeding [45,57,58]. From a behavioural viewpoint, heterodonty in extant and extinct reptiles has been

suggested to have enabled consumption of larger ranges of food items through preferential tooth use for piercing, handling or crushing different items [39,45,51,57,59,60]. This has potentially severe implications for DMTA sampling protocols as different teeth in the tooth row may contain different dietary signals. Determining whether microwear differences can be explained by tooth shape in addition to, or instead of, diet will provide a more thorough understanding of how taxa with non-occlusal dentitions interact with food items. The lack of homologous landmarks on non-occlusal teeth, however, can limit the power of sophisticated techniques that quantify tooth shape, such as Euclidean measurements [39]. Alternatively, tooth aspect ratio (maximum height/maximum width) does not rely on homologous landmarks and has been shown to be a robust measure for understanding changes in tooth pressure experiments [61]. Aspect ratio is therefore a simple yet informative proxy for understanding the influence of tooth shape on intraspecific microwear texture differences.

This study presents exploratory quantitative analyses of three-dimensional tooth microwear textures from non-occlusal surfaces along the tooth rows of extant reptiles, and on the influence of endogenous non-dietary variables on intraspecific microwear texture differences along tooth rows. Data from six crocodilian (Crocodylia) and seven varanid (Varanidae) species with well-constrained diets were used to test the null hypothesis, for each species, that microwear textures do not differ between teeth from different positions along the tooth row. Data from species where this null hypothesis could be rejected were then used to test the subsequent null hypotheses that tooth microwear texture differences from different positions of the tooth row do not correlate with the following non-dietary variables: (i) tooth position in the tooth row, (ii) mechanical advantage, and (iii) tooth aspect ratio. We then used the multivariate dietary framework of [10] to test the final null hypothesis (again using data from species where the first null hypothesis could be rejected), that reptile tooth microwear textures from different positions of the tooth row do not occupy different areas of a texture–dietary space comprising extant reptiles with well-constrained diets.

# 2. Material and methods

## 2.1. Species used and material sampled

Tooth microwear textures were sampled from six species of crocodilian and seven species of varanid: *Alligator mississippiensis* ($n = 10$); *Caiman crocodilus* ($n = 8$); *Crocodylus acutus* ($n = 8$); *Crocodylus niloticus* ($n = 5$); *Crocodylus porosus* adults ($n = 8$); *Crocodylus porosus* juveniles ($n = 7$); *Gavialis gangeticus* ($n = 9$); *Varanus komodoensis* ($n = 7$); *Varanus nebulosus* ($n = 11$); *Varanus niloticus* ($n = 8$); *Varanus olivaceus* ($n = 8$); *Varanus prasinus* ($n = 7$); *Varanus rudicollis* ($n = 8$) and *Varanus salvator* ($n = 10$). Specimens were sampled from the Field Museum of Natural History, Chicago, Illinois, USA (FMNH); Grant Museum of Zoology, University College London, London, UK (LDUCZ); Natural History Museum, London, UK (NHMUK); University of Oxford Museum of Natural History, Oxford, UK (OUMNH); Florida Museum of Natural History, Gainesville, Florida, USA (UF) and the National Museum of Natural History, Smithsonian Institute, Washington D.C., USA (USNM). Full details of sampled specimens are available in the electronic supplementary material, table S1. Schematic diagrams of reptile mandibles were traced or drawn from the following sources: *A. mississippiensis*, [45] under a Creative Commons Attribution open-access license; *Ca. crocodilus*, FMNH 73700; *Cr. acutus*, *G. gangeticus* and *V. rudicollis*, all [27] under a Creative Commons Attribution 4.0 International License (https://creativecommons.org/licenses/by/4.0/).

The availability of museum specimens allowed multiple *Cr. porosus* life-history stages to be included since the tooth morphology of some crocodilians, including *Cr. porosus*, changes with ontogeny [40,57,62]. In younger specimens, all teeth in the tooth row are relatively slender and sharp, but as individuals age, the distally positioned teeth become rounder and more robust [40,57,62]. Specimens with lower jaw lengths of less than 50 cm were classified as juveniles; specimens with jaw lengths exceeding 50 cm were considered to be adults [63].

Information on diets was taken from electronic supplementary material, table S1 from [10]. In the current study, dietary information was not needed for microwear analysis but was required for informing discussions based on the identified microwear texture differences (or lack of texture differences) along reptile tooth rows.

## 2.2. Microwear sampling strategy

Microwear surface texture data were acquired from the mandible teeth of dry skeletal specimens. Mandibles were theoretically divided into five regions to test the hypothesis that microwear textures

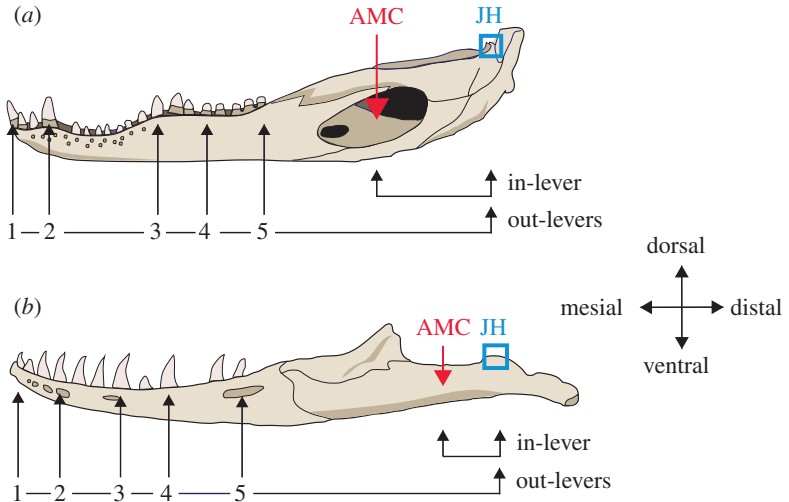

**Figure 1.** Schematic illustrating microwear sampling strategy and mechanical advantage (in-lever distance/out-lever distance) along the mandible tooth row, exemplified by: (*a*) *Alligator mississippiensis* and (*b*) *Varanus rudicollis*. In-lever distance measures from the jaw hinge (JH; medial dorsal point of the articular) to the central insertion point of the jaw adductor muscles (AMC). Out-lever distances measure from the jaw hinge to sampled teeth. Out-levers 1–5 denote teeth from the mesial, mesial–middle, middle, middle–distal and distal areas of the tooth row, respectively. In-lever, out-lever and mechanical advantage data are included in the electronic supplementary material, table S1. Diagrams not to scale. The *A. mississippiensis* mandible was redrawn from [45] under a Creative Commons Attribution open-access license and the *V. rudicollis* mandible was redrawn from [27] under a Creative Commons Attribution 4.0 International License (https://creativecommons.org/licenses/by/4.0/).

differ between teeth from different positions of the tooth row. These regions denote the mesial, mesial–middle, middle, middle–distal and distal regions of the mandible tooth row and sampled teeth from these regions are termed out-levers (OL) 1–5, respectively (figure 1). The underlying assumption of these regions is that in all sampled taxa, the mesially positioned teeth will be subjected to higher bite forces and the distally positioned teeth will be subjected to lower forces (see §2.4). Given the substantial differences in body size and bite force between the study reptiles, comparisons of the relative forces experienced by teeth can only be done meaningfully within the tooth row of an individual. As only the teeth are being analysed, and the extent of the tooth row along the mandible will vary between taxa, the mesial and distal limits for the five loading regions will also vary between taxa. However, as wear differences between teeth from different positions within the same individual are the prime focus of this study, the absolute differences in the positions of the five regions does not affect the final analysis. The most fully erupted tooth from each out-lever position was sampled, as these teeth would have more likely come into contact with food items and for easier data collection. Non-occlusal labial surfaces, as close to the apex as possible, were sampled for all teeth. Wear facets that probably formed from tooth–tooth occlusion from the opening and closing of jaws, characterized by their vertical orientation, elliptical shape and parallel features [38,64], were not sampled. Broken teeth that had more than a quarter of their estimated original height missing were not sampled to increase the likelihood that all sampled tooth apexes had experienced regular tooth–food contact. No preference was given to right or left sides of mandibles and teeth from both sides were pooled together for analysis. Imbalanced sample sizes occur between out-levers and between species for several reasons, including specimen availability in museums and missing teeth in specimens. See the electronic supplementary material, table S1 for a full breakdown of which out-levers were sampled from each specimen.

Teeth were cleaned using 70% ethanol-soaked cotton swabs to remove dirt and consolidant. High fidelity moulds were taken of teeth using President Jet Regular Body polyvinylsiloxane (Coltène/Whaledent Ltd, Burgess Hill, West Sussex, UK). Initial moulds taken from each tooth were discarded to remove any remaining dirt, with all analyses performed on the second moulds. Casts were made from these moulds using EpoTek 320 LV Black epoxy resin mixed to manufacturer's instructions. The resin was cured for 24 h under 2 bar/30 psi of pressure (Protima Pressure Tank 10 l) to improve casting quality. Small casts were mounted onto 12.7 mm SEM stubs using President Jet polyvinylsiloxane with the labial, non-occluding surfaces orientated dorsally to optimize data acquisition. All casts were sputter coated with gold for 3 min (SC650, Bio-Rad, Hercules, CA, USA) to optimize the capture of surface texture data. Replicas produced using these methods are statistically indistinguishable from original tooth surfaces [65].

## 2.3. Surface texture data acquisition

Surface texture data acquisition follows standard laboratory protocols [10]. Data were captured using an Alicona Infinite Focus microscope G4b (IFM; Alicona GmbH, Graz, Austria; software version 5.1), using a × 100 objective lens, producing a field of view of 146 × 100 µm. Lateral and vertical resolution were set at 440 and 20 nm respectively. Casts were orientated so labial surfaces were perpendicular to the axis of the objective lens.

All three-dimensional data files were processed using the Alicona IFM software (version 5.1) to remove dirt particles from tooth surfaces and anomalous data points (spikes) by manual deletion. Data were levelled (subtraction of least-squares plane) to remove variation caused by differences in tooth surface orientation at the time of data capture. Files were exported as .sur files and imported into Surfstand (software version 5.0.0 Centre for Precision Technologies, University of Huddersfield, West Yorkshire, UK). Scale-limited surfaces were generated through the application of a fifth-order robust polynomial to remove gross tooth form and a robust Gaussian filter (wavelength $\lambda_c = 0.025$ mm) [6,9]. International Organization for Standardization (ISO) 25178-2 texture parameters [3] were then generated from each scale-limited surface. Brief ISO parameter definitions can be found in table 1. Further details can be found in the electronic supplementary material table S2 and electronic supplementary material, figure S2, both from [10].

## 2.4. Mechanical advantage calculation

To explore whether microwear along the tooth row varies with bite force, a two-dimensional model was used to calculate mechanical advantage at sampled areas along the mandible tooth row, assuming a simple lever for the opening and closing of the mandible. The fulcrum of the lower jaw is the jaw hinge, i.e. the medial dorsal point on the articular (figure 1). The mandible, plus a theoretical adductor muscle pulling the jaw closed, is viewed as a third-class lever, where the force exerted by muscles is located in between the fulcrum and the biting position [66]. Mechanical advantage was calculated using the equation

$$\text{Force.Out} = \frac{\text{Force.In} \times \text{in-lever}}{\text{out-lever}},$$

where Force.Out denotes experienced bite force at the site of study in the mandible, Force.In denotes applied muscle force, in-lever is the horizontal distance from the jaw-joint to the mid-point of the adductor musculature and out-lever is the distance from the jaw-joint to the tooth of interest [49,50].

To facilitate mechanical advantage comparisons between taxa, Force.In was set at 1.0, which results in Force.Out becoming a fraction of the total force in [49,50,53]. This also simplifies the study of skeletal specimens because biological structures needed for theoretically calculating actual bite forces, such as the adductor musculature, are usually absent [31,42,49,53].

In-levers were measured as the horizontal distance from the jaw hinge to the middle of the masseter attachment point, which was parallel to the flat surface upon which the ventral surface of specimens were laid (figure 1). Out-levers were similarly measured as horizontal distances from the jaw hinge to the non-occlusal surfaces of respective sampled teeth, parallel to the ventral surface of specimens resting upon a flat surface (figure 1) (all measurements in millimetres). Distances in most specimens were measured using Precision Gold digital callipers to the nearest 0.01 mm except for the out-levers from the largest crocodilian specimens which were measured using a tape measure to the nearest millimetre.

It should be noted that this two-dimensional lever model does not take into account adductor musculature insertion angles, which can result in overestimations of mechanical advantage [42,44,67,68]. This is not a problem for this study since effects of absolute bite force on microwear patterns are not tested for. In-lever, out-lever and mechanical advantage measurements for species that exhibited microwear differences along the tooth row are included in the electronic supplementary material, table S1.

## 2.5. Tooth shape

To explore whether microwear differences between teeth from different positions of the tooth row vary with tooth shape, the aspect ratio of sampled teeth was calculated. Aspect ratio is a simple, yet relatively robust, two-dimensional measure for quantifying tooth shape [61], since crocodilians and varanids have monocuspid teeth with few homologous landmarks [39,40,56]. Aspect ratio was calculated from labial viewpoints by dividing tooth height—the apex to the crown enamel base or to the perpendicular

**Table 1.** Definition, description, and categorization of the 21 International Organization for Standardization three-dimensional texture parameters used in this study. Many parameters are derived from the areal material ratio curve; a cumulative probability density function derived from the scale-limited tooth surface by plotting the cumulative percentage of the tooth surface against height. The peaks, valleys and core material of tooth surfaces are defined on the basis of this curve, with the core for material ratio parameters equivalent to the volume that lies between the heights of the surface delimited by the extrapolated intercept of the minimum slope of the curve. See electronic supplementary material, table S2 and electronic supplementary material, figure S2 both from [10] for more detailed parameter definitions and for more information on how parameter values are derived from surface textures, respectively.

| parameter | unit | definition | category |
|---|---|---|---|
| Sq | µm | Root-mean-square height of surface | height |
| Sp | µm | Maximum peak height of surface. Based on only one peak | height |
| Sv | µm | Maximum valley depth of surface. Based on only one valley | height |
| Sz | µm | Maximum height of surface, calculated by subtracting the maximum valley depth from peak height | height |
| Sa | µm | Average height of the surface texture | height |
| Sku | — | Kurtosis of height distribution of surface | height |
| S5z | µm | 10 point height of surface, average value of the five highest peaks and the five deepest valleys | feature |
| Sdq | — | Root-mean-square gradient of the surface | hybrid |
| Sdr | % | Developed interfacial area ratio | hybrid |
| Sds | $mm^{-2}$ | Density of summits. Number of summits per unit area making up the surface | hybrid |
| Ssc | $µm^{-1}$ | Mean summit curvature for peak structures | |
| Sk | µm | Core roughness depth, height of the core material | material ratio |
| Spk | µm | Mean height of the peaks above the core material | material ratio |
| Svk | µm | Mean depth of the valleys below the core material | material ratio |
| Smr1 | % | Surface bearing area ratio (proportion of the surface which consists of peaks above the core material) | material ratio |
| Smr2 | % | Surface bearing area ratio (proportion of the surface which would carry the load) | material ratio |
| Vmp | $µm^3 mm^{-2}$ | Material volume of the peaks of the surface | volume |
| Vmc | $µm^3 mm^{-2}$ | Material volume of the core of the surface | volume |
| Vvc | $µm^3 mm^{-2}$ | Void volume of the core of the surface | volume |
| Vvv | $µm^3 mm^{-2}$ | Void volume of the valleys of the surface | volume |
| Str | — | Texture aspect ratio | spatial |

junction between the visible crown and host bone—by maximum tooth width in a mesio-distal direction [61]. Tooth height, width and aspect ratio data for species that exhibited microwear differences along the tooth row are included in the electronic supplementary material, table S1.

## 2.6. Statistical analyses

Log-transformed texture data were used for analyses as some of the texture parameters were non-normally distributed (Shapiro–Wilk, $p > 0.05$). The parameter Ssk—skewness of the height distribution of the three-dimensional surface texture (table 1)—was excluded from analyses as it contains negative values and thus cannot be log-transformed.

To test the null hypothesis that microwear textures do not differ between teeth from different positions along the tooth row, analysis of variance (ANOVA) with pairwise testing (Tukey HSD) was independently applied to each texture parameter for each species. Where homogeneity of variance

tests (Bartlett and Levene tests) revealed evidence of unequal variances, Welch ANOVA is reported. Additional analyses were employed in reptiles which exhibited significant textural differences, to further test the relationships between texture and out-lever position. In each species, average parameter values of each parameter were calculated and the average values of each parameter were separately ranked between out-lever positions from most to least positive. Matched pairs *t*-tests were independently used to compare the profiles of average parameter values between out-lever positions for each species. Out-lever number was treated as a fixed factor for these tests.

For reptile species where the first null hypothesis was rejected, Spearman's rank was used to test for correlations between texture parameters that exhibited significant differences along the tooth row of that species and the non-dietary variables: (i) out-lever number, (ii) mechanical advantage, and (iii) tooth aspect ratio. In each species, the out-lever number was first correlated with aspect ratio to quantify the relationship of tooth shape along the tooth row. The out-lever number and mechanical advantage were not correlated with each other because the latter naturally increases as the distance to the fulcrum decreases due to the function of its calculation. Correlations were then performed independently in each species between their significant texture parameters and non-dietary variables. The out-lever number was treated as a covariate in these tests. To test the final null hypothesis that tooth microwear textures from different positions of the tooth row do not occupy different areas of texture–dietary space, the OL2–5 texture data of species where the first null hypothesis was rejected were independently projected into the first two principal component axes of the reptile multivariate space from [10]. This texture-space was constructed using four ISO texture parameters that significantly differed between the OL1 teeth of the reptile dietary guilds (Spk, mean height of peaks above the core material; Sds, summit density; Vmp, material volume of peaks; and Smr1, proportion of surface that consists of peaks). The OL1 data were not projected in order to avoid pseudoreplication, as this texture data comprises part of the space. ANOVA with pairwise testing (Tukey HSD) was independently applied to each species to test whether out-levers occupied different areas of multivariate space.

A Benjamini–Hochberg (B–H) procedure was used to account for the possibility of inflated type I error rates associated with multiple comparisons [69]. The false discovery rate was set at 0.05. The B–H procedure was not needed for the Tukey HSD tests as it already accounts for inflated type I error rates [34].

All analyses were performed with JMP Pro 12 (SAS Institute, Cary, NC, USA) except for the B–H procedure, which used Microsoft Excel ([70] www.biostathandbook.com/multiplecomparisons.html).

# 3. Results

## 3.1. Microwear textural differences between tooth row positions

Tooth surface textures do not differ along the tooth row in eight reptile species (*Cr. niloticus*, *Cr. porosus* adults, *Cr. porosus* juveniles, *V. komodoensis*, *V. nebulosus*, *V. niloticus*, *V. olivaceus*, *V. prasinus* and *V. salvator*; all texture parameters $p > 0.05$, electronic supplementary material, tables S2–S10 respectively). The first null hypothesis, therefore, could not be rejected in these reptiles. Tooth surface textures significantly differ along the tooth row for at least one parameter in five reptiles (*A. mississippiensis*, *Ca. crocodilus*, *Cr. acutus*, *G. gangeticus* and *V. rudicollis*; all significant parameters $p < 0.05$, tables 2–6, respectively). The first null hypothesis was therefore rejected in these reptiles.

Ten parameters significantly differ along the *A. mississippiensis* tooth row: Sq (root-mean-square height of surface), Sp (maximum peak height), Sz (maximum surface height), Vmp, Vmc (material volume of core), Vvc (void volume of core), Spk, Sk (core roughness depth), S5z (10 point height of surface) and Sa (average surface height; tables 1 and 2). Tukey HSD pairwise testing indicates that out-lever 1 (OL1) teeth significantly differ from OL2 teeth for five parameters (lower Sp, Vmp, Vvc, Spk and S5z) but do not differ from OL3, OL4 or OL5 teeth (figures 2 and 3). OL2 teeth differ from OL4 teeth for 10 parameters (lower Sq, Sp, Sz, Vmp, Vmc, Vvc, Spk, Sk, S5z and Sa) and from OL5 teeth for two parameters (S5z and Sa), but do not differ from OL3 teeth (table 2; figures 2 and 3). OL4 teeth therefore have the roughest microwear textures, followed by OL5 teeth, and OL2 teeth have the smoothest textures, followed by OL1 teeth. Matched pairs *t*-tests show that all out-levers significantly differ from each other except for out-levers 4 and 5 (electronic supplementary material, table S11). OL4 textures exhibit the tallest and largest peaks, the highest surface elevation and the highest material and void volume of the core. OL5 textures exhibit the largest and deepest valleys. Alternatively, OL2 textures exhibit the lowest and smallest peaks, the shallowest valleys, and the

**Table 2.** ANOVA results (4 decimal places (d.p.)) of ISO texture parameters between teeth from different positions of the tooth row in *Alligator mississippiensis*. Data log-transformed and scale-limited using a fifth-order polynomial and a robust Gaussian filter. Includes results of Benjamini–Hochberg (B–H) procedure; texture parameters exhibiting significant differences after B–H application shown in italics.

| parameter | F-ratio | p-value | d.f. |
|---|---|---|---|
| *Sq* | *5.4971* | *0.0016* | *4, 34* |
| Sku | 1.1937 | 0.3313 | 4, 34 |
| *Sp* | *4.4126* | *0.0056* | *4, 34* |
| Sv | 1.2316 | 0.3159 | 4, 34 |
| *Sz* | *3.6702* | *0.0137* | *4, 34* |
| Sds | 1.1434 | 0.3529 | 4, 34 |
| Str | 2.3575 | 0.0731 | 4, 34 |
| Sdq | 2.5861 | 0.0543 | 4, 34 |
| Ssc | 2.4572 | 0.0642 | 4, 34 |
| Sdr | 2.6536 | 0.0498 | 4, 34 |
| *Vmp* | *6.1878* | *0.0007* | *4, 34* |
| *Vmc*[a] | *11.6462* | *0.0002* | *4, 14.956* |
| *Vvc*[a] | *16.1992* | *<0.0001* | *4, 14.451* |
| Vvv | 2.1162 | 0.1002 | 4, 34 |
| *Spk* | *6.3111* | *0.0007* | *4, 34* |
| *Sk*[a] | *8.4077* | *0.001* | *4, 14.706* |
| Svk | 1.6555 | 0.1831 | 4, 34 |
| Smr1 | 2.4773 | 0.0625 | 4, 34 |
| Smr2 | 1.4161 | 0.2497 | 4, 34 |
| *S5z* | *7.539* | *0.0002* | *4, 34* |
| *Sa* | *6.1918* | *0.0007* | *4, 34* |

[a]Indicates Welch Test result (ANOVA of unequal variances, Bartlett and/or Levene test).

lowest surface elevation. This strengthens the conclusions that OL4 teeth have the roughest textures and that OL2 teeth have the smoothest textures, followed by OL1 textures. OL3 textures have the highest peak density and load-bearing ratio. See electronic supplementary material, table S12 for average parameter value rankings between all out-lever positions for each texture parameter.

Eight parameters significantly differ along the *Ca. crocodilus* tooth row: Sq, Sdq (root-mean-square gradient of surface), Sdr (developed interface ratio), Vmc, Vvc, Sk, Svk (mean depth of valleys below the core) and Sa (tables 1 and 3). Tukey HSD pairwise testing indicates that OL1 teeth differ from OL4 teeth for six parameters (lower Sdq, Sdr, Vmc, Vvv, Sk and Sa), from OL2 and OL5 teeth for three parameters each (lower Sdq, Sdr and Sk in both cases), but do not differ from OL3 teeth (table 3; figures 4 and 5). OL4 teeth therefore have the roughest microwear textures, followed by OL5 and OL2 teeth, and OL1 teeth have the smoothest textures. Matched pairs *t*-tests show that all out-levers significantly differ from each other, except for out-levers 2 and 3, and 4 and 5 (electronic supplementary material, table S13). OL4 textures have the tallest and largest peaks, the highest surface elevation, the highest material and void volume of the core and the largest and deepest valleys. OL5 textures have the greatest peak density and the highest proportion of the surface area that consists of peaks. By contrast, OL1 textures have the lowest surface elevation, the lowest and smallest peaks and the smallest (by volume) and shallowest valleys. This further indicates that OL4 teeth have the roughest textures, followed by OL5 teeth, and that OL1 teeth have the smoothest textures. See electronic supplementary material, table S14 for average parameter value rankings.

One parameter significantly differs along the *Cr. acutus* tooth row: Ssc (mean peak curvature; tables 1 and 4). Tukey HSD pairwise testing indicates that OL1 teeth differ from OL4 and OL5 teeth (lower Ssc in both cases; table 4; figures 6 and 7). OL4 and OL5 textures therefore have the most curved textural peaks,

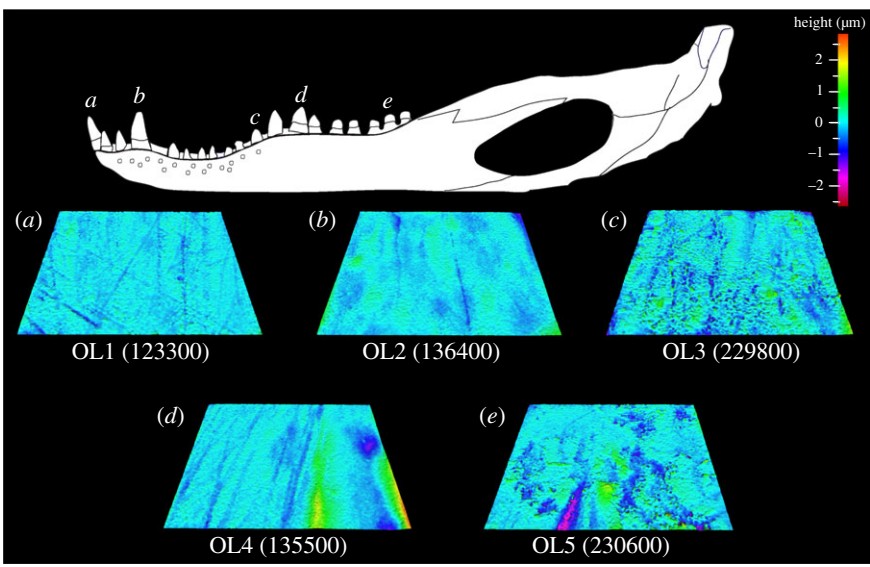

**Figure 2.** Example scale-limited tooth surfaces along the *Alligator mississippiensis* tooth row. (*a*) mesial; (*b*) mesial–middle; (*c*) middle; (*d*) middle–distal and (*e*) distal. Topographic scale in micrometres. Example surfaces are illustrative and do not all originate from the same individual. Numbers within brackets denote Leicester IFM sample numbers. The *A. mississippiensis* mandible was redrawn from fig. 1 of [45] under a Creative Commons Attribution open-access license.

**Table 3.** ANOVA results (4 d.p.) of ISO texture parameters between teeth from different positions of the tooth row in *Caiman crocodilus*. Data log-transformed and scale-limited using a fifth-order polynomial and a robust Gaussian filter. Includes results of Benjamini–Hochberg (B–H) procedure; texture parameters exhibiting significant differences after B–H application shown in italics.

| parameter | *F*-ratio | *p*-value | d.f. |
|---|---|---|---|
| *Sq*[a] | *10.9052* | *0.0003* | *4, 14.804* |
| Sku | 0.808 | 0.5296 | 4, 31 |
| Sp | 0.9559 | 0.4454 | 4, 31 |
| Sv | 1.4233 | 0.2495 | 4, 31 |
| Sz | 1.5625 | 0.2089 | 4, 31 |
| Sds | 0.2601 | 0.9012 | 4, 31 |
| Str | 2.2854 | 0.0825 | 4, 31 |
| *Sdq* | *4.0096* | *0.0098* | *4, 31* |
| Ssc | 2.9042 | 0.0376 | 4, 31 |
| *Sdr* | *4.1612* | *0.0082* | *4, 31* |
| Vmp | 1.0052 | 0.4198 | 4, 31 |
| *Vmc*[a] | *9.6875* | *0.0004* | *4, 14.999* |
| *Vvc*[a] | *8.6085* | *0.0008* | *4, 15.058* |
| *Vvv*[a] | 3.8396 | 0.0238 | 4, 15.247 |
| Spk | 1.0029 | 0.4209 | 4, 31 |
| *Sk*[a] | *9.4135* | *0.0005* | *4, 15.181* |
| *Svk*[a] | *4.2268* | *0.017* | *4, 15.206* |
| Smr1 | 0.8386 | 0.5113 | 4, 31 |
| Smr2 | 0.1069 | 0.9792 | 4, 31 |
| S5z | 1.3888 | 0.2607 | 4, 31 |
| *Sa*[a] | *11.3546* | *0.0002* | *4, 15.072* |

[a]Indicates Welch Test result (ANOVA of unequal variances, Bartlett and/or Levene test).

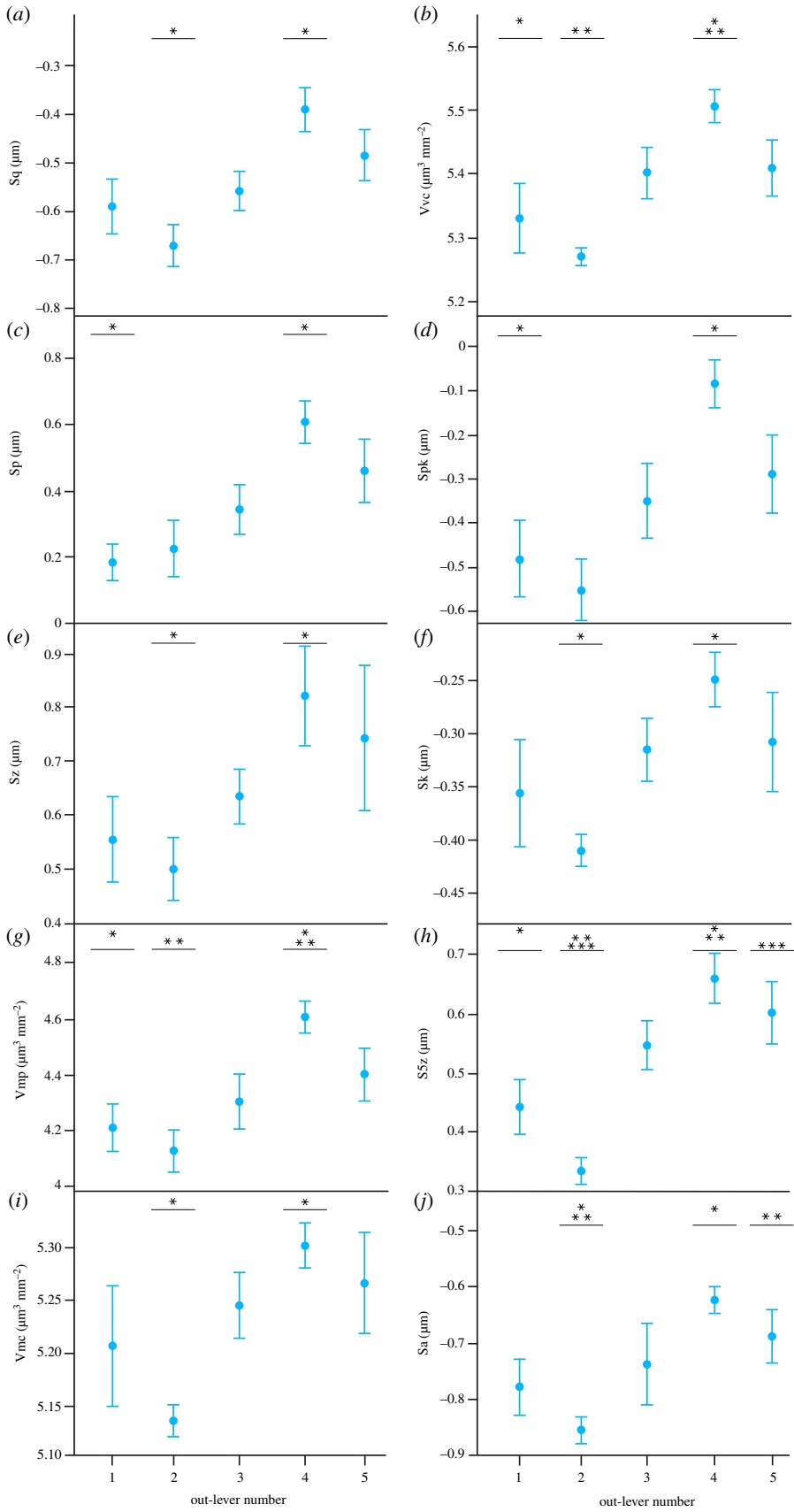

**Figure 3.** Microwear textural differences (mean ± s.e.) along the *Alligator mississippiensis* tooth row for each ISO parameter that significantly differs between tooth positions (ANOVA). Out-lever numbers 1–5 denote mesial, mesial–middle, middle, middle–distal and distal tooth positions, respectively. Asterisks represent species-level significant Tukey HSD pairwise differences between out-levers for each parameter.

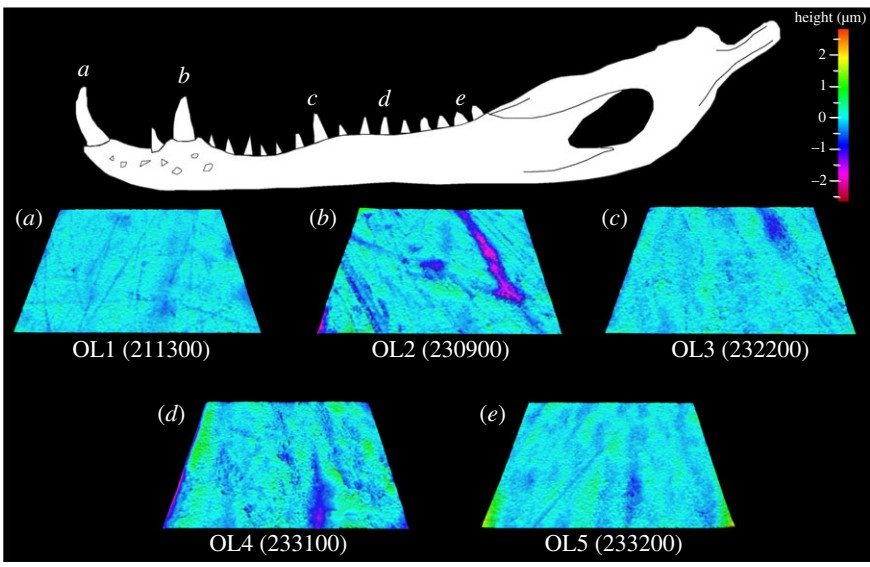

**Figure 4.** Example scale-limited tooth surfaces along the *Caiman crocodilus* tooth row. (*a*) mesial; (*b*) mesial–middle; (*c*) middle; (*d*) middle–distal and (*e*) distal. Topographic scale in micrometres. Example surfaces are illustrative and do not all originate from the same individual. Numbers within brackets denote Leicester IFM sample numbers. The *Ca. crocodilus* mandible was drawn from specimen FMNH 73700.

**Table 4.** ANOVA results (4 d.p.) of ISO texture parameters between teeth from different positions of the tooth row in *Crocodylus acutus*. Data log-transformed and scale-limited using a fifth-order polynomial and a robust Gaussian filter. Includes results of Benjamini–Hochberg (B–H) procedure; texture parameters exhibiting significant differences after B–H application shown in italics.

| parameter | F-ratio | p-value | d.f. |
|---|---|---|---|
| Sq | 1.3765 | 0.2744 | 4, 22 |
| Sku | 0.7334 | 0.579 | 4, 22 |
| Sp | 0.7108 | 0.5933 | 4, 22 |
| Sv | 0.2909 | 0.8807 | 4, 22 |
| Sz | 0.6885 | 0.6076 | 4, 22 |
| Sds | 1.5797 | 0.2149 | 4, 22 |
| Str | 0.551 | 0.7003 | 4, 22 |
| Sdq | 1.4037 | 0.2656 | 4, 22 |
| *Ssc* | *5.997* | *0.002* | *4, 22* |
| Sdr | 1.5662 | 0.2184 | 4, 22 |
| Vmp | 1.7626 | 0.1724 | 4, 22 |
| Vmc | 0.648 | 0.6342 | 4, 22 |
| Vvc | 1.0919 | 0.385 | 4, 22 |
| Vvv | 1.3897 | 0.2701 | 4, 22 |
| Spk | 1.5862 | 0.2132 | 4, 22 |
| Sk | 0.7965 | 0.5402 | 4, 22 |
| Svk | 1.2339 | 0.3254 | 4, 22 |
| Smr1 | 0.7626 | 0.5608 | 4, 22 |
| Smr2 | 0.98 | 0.4387 | 4, 22 |
| S5z | 0.4893 | 0.7435 | 4, 22 |
| Sa | 1.2237 | 0.3294 | 4, 22 |

**Table 5.** ANOVA results (4 d.p.) of ISO texture parameters between teeth from different positions of the tooth row in *Gavialis gangeticus*. Data log-transformed and scale-limited using a fifth-order polynomial and a robust Gaussian filter. Includes results of Benjamini–Hochberg (B–H) procedure; texture parameters exhibiting significant differences after B–H application shown in italics.

| parameter | F-ratio | p-value | d.f. |
|---|---|---|---|
| Sq[a] | 3.845 | 0.0217 | 4, 16.552 |
| Sku | 0.9096 | 0.4694 | 4, 34 |
| Sp | 2.2112 | 0.0885 | 4, 34 |
| Sv | 0.8464 | 0.5057 | 4, 34 |
| Sz | 0.897 | 0.047 | 4, 34 |
| Sds | 3.6855 | 0.0263 | 4, 34 |
| Str | 0.3328 | 0.8539 | 4, 34 |
| Sdq | 1.3255 | 0.2804 | 4, 34 |
| Ssc | 1.3057 | 0.2875 | 4, 34 |
| Sdr | 1.3408 | 0.2749 | 4, 34 |
| Vmp | 0.6789 | 0.6114 | 4, 34 |
| Vmc | 3.4305 | 0.0185 | 4, 34 |
| Vvc | 2.4871 | 0.0618 | 4, 34 |
| *Vvv*[a] | *6.0898* | *0.0034* | *4, 16.379* |
| Spk | 0.8881 | 0.4815 | 4, 34 |
| Sk | 3.0384 | 0.0303 | 4, 34 |
| *Svk*[a] | *5.4732* | *0.0054* | *4, 16.4* |
| *Smr1* | *3.8516* | *0.011* | *4, 34* |
| *Smr2*[a] | *5.1492* | *0.0081* | *4, 15.111* |
| S5z | 1.4237 | 0.2472 | 4, 34 |
| *Sa*[a] | *4.8409* | *0.009* | *4, 16.492* |

[a]Indicates Welch Test result (ANOVA of unequal variances, Bartlett and/or Levene test).

and OL1 textures have the least curved peaks. Matched pairs *t*-tests show that all out-levers significantly differ from each other except for: (i) 1 and 3, (ii) 1 and 4, and (iii) 3 and 4 (electronic supplementary material, table S15). OL5 textures have the highest surface elevation for the whole tooth surface and the core material, the tallest and largest peaks and highest void and material volume of the core. OL4 textures have the most curved peaks and the highest elevational gradients. OL1 textures have the largest (by volume) and deepest valleys. OL5 textures have the lowest surface elevation, lowest and smallest peaks and void volume of the core. This alternatively indicates that OL5 teeth have the roughest textures, OL4 and OL1 teeth have similarly rough textures and OL2 teeth have the smoothest textures. See electronic supplementary material, table S16 for average parameter value rankings between all out-lever positions for each texture parameter.

Five parameters significantly differ along the *G. gangeticus* tooth row: Vvv (void volume of valleys), Svk, Smr1, Smr2 (surface bearing area ratio) and Sa (tables 1 and 5). Tukey HSD pairwise testing indicates that OL1 teeth differ from OL3 teeth for four parameters (higher Vvv and Sa, lower Smr1 and Smr2), from OL4 teeth for two parameters (lower Smr1 and Smr2), from OL5 teeth for two parameters (higher Vvv, lower Smr2), but do not differ from OL2 teeth (table 5; figures 8 and 9). Microwear textures along the tooth row can therefore be considered rougher or smoother with respect to different textural characteristics. OL1 teeth for example have the highest average surface elevation and largest valleys (by volume). OL3, OL4 and, to a lesser extent, OL5 teeth have the highest percentages of tooth surface areas that consist of peaks and that are load-bearing surfaces. Matched pairs *t*-tests show that all out-levers significantly differ from each other except for: (i) 1 and 2, (ii) 1 and 4, and (iii) 3 and 4 (electronic supplementary material, table S17). OL4 textures have the largest peaks (by volume) that comprise more of the surface texture area, and the highest void volume of the core material. OL1 textures, and to a lesser extent OL2 textures, have the highest average surface

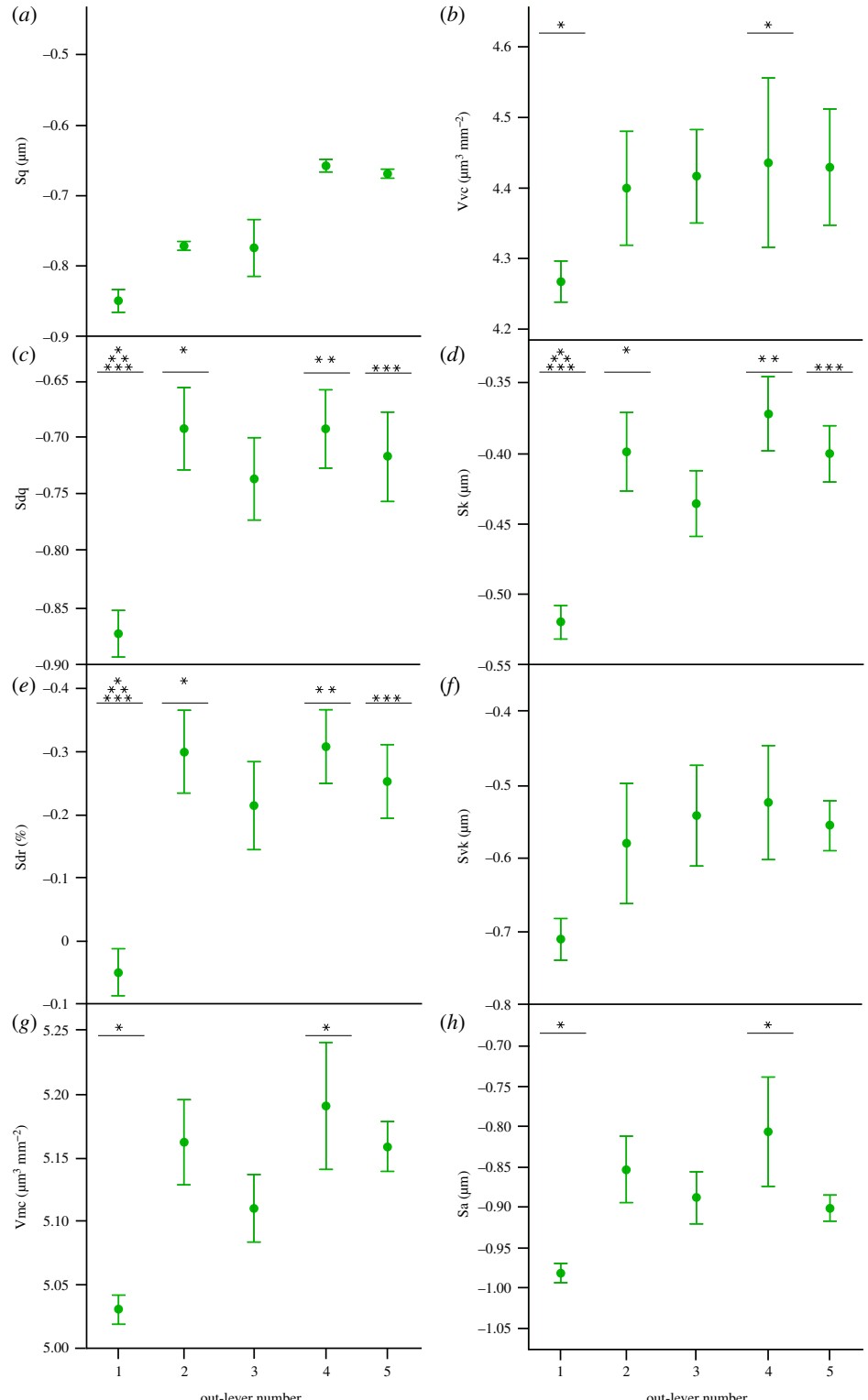

**Figure 5.** Microwear textural differences (mean ± s.e.) along the *Caiman crocodilus* tooth row for each ISO parameter that significantly differs between tooth positions (ANOVA). Out-lever numbers 1–5 denote mesial, mesial–middle, middle, middle–distal and distal tooth positions, respectively. Asterisks represent species-level significant Tukey HSD pairwise differences between out-levers for each parameter. N.B. Parameters Sp and Svk exhibit significant differences between out-levers (Welch ANOVA) but no significant pairwise differences.

elevation and the deepest and largest (by volume) valleys. OL5 textures have the smallest peaks (by volume) and shallowest valleys. Overall, and somewhat contradictory to the ANOVA results, OL4 teeth have the roughest textures, OL1 and OL2 teeth exhibit similar, somewhat rough textures and

**Table 6.** ANOVA results (4 d.p.) of ISO texture parameters between teeth from different positions of the tooth row in *Varanus rudicollis*. Data log-transformed and scale-limited using a fifth-order polynomial and a robust Gaussian filter. Includes results of Benjamini–Hochberg (B–H) procedure; texture parameters exhibiting significant differences after B–H application shown in italics.

| parameter | F-ratio | p-value | d.f. |
|---|---|---|---|
| *Sq*[a] | *5.7167* | *0.0071* | *4, 12.885* |
| Sku | 0.3352 | 0.8519 | 4, 28 |
| Sp | 2.8071 | 0.0446 | 4, 28 |
| Sv | 3.8784 | 0.0323 | 4, 28 |
| Sz | 2.7522 | 0.0477 | 4, 28 |
| Sds | 1.4844 | 0.2336 | 4, 28 |
| Str | 0.997 | 0.4256 | 4, 28 |
| Sdq | 0.8067 | 0.5314 | 4, 28 |
| Ssc | 0.9953 | 0.4265 | 4, 28 |
| Sdr | 0.8602 | 0.4998 | 4, 28 |
| *Vmp* | *4.229* | *0.0085* | *4, 28* |
| *Vmc* | *5.202* | *0.0029* | *4, 28* |
| *Vvc* | *5.5045* | *0.0021* | *4, 28* |
| Vvv | 2.1661 | 0.0989 | 4, 28 |
| *Spk* | *4.7416* | *0.0048* | *4, 28* |
| *Sk* | *6.0232* | *0.0013* | *4, 28* |
| Svk | 2.3032 | 0.0833 | 4, 28 |
| Smr1 | 0.0999 | 0.9816 | 4, 28 |
| Smr2 | 1.7243 | 0.1726 | 4, 28 |
| *S5z*[a] | *5.1565* | *0.0119* | *4, 11.972* |
| *Sa*[a] | *5.1159* | *0.0102* | *4, 13.381* |

[a]Indicates Welch Test result (ANOVA of unequal variances, Bartlett and/or Levene test).

OL5 teeth have the smoothest textures. See electronic supplementary material, table S18 for average parameter value rankings between all out-lever positions for each texture parameter.

Eight parameters significantly differ along the *V. rudicollis* tooth row: Sq, Vmp, Vmc, Vvc, Spk, Sk, S5z and Sa (tables 1 and 6). Tukey HSD pairwise testing indicates that OL1 teeth differ from OL3, OL4 and OL5 teeth for five parameters (higher Vmp, Vmc, Vvc, Spk and Sk in all cases), but do not differ from OL2 teeth (table 6; figures 10 and 11). OL1 teeth therefore have the roughest microwear textures and OLs 3–5 have smoother textures. Matched pairs *t*-tests show that all out-levers significantly differ from each other except for 3 and 5 (electronic supplementary material, table S19). OL1 textures have the highest average surface elevation and elevational gradient, the highest and largest peaks and largest material and void volume of the core. OL2 textures have the deepest and largest (by volume) valleys. OL4 textures have the lowest average surface elevation and lowest material and void volume of the core. This strengthens the conclusion that OL1 textures have the roughest microwear textures, followed by OL2 textures, and also indicates that OL4 teeth have the smoothest textures, with OL3 and OL5 tooth textures very similar to each other. See electronic supplementary material, table S20 for average parameter value rankings between all out-lever positions for each texture parameter.

## 3.2. Correlations between microwear textural differences and non-dietary variables

Out-lever number strongly negatively correlates with tooth position in *A. mississippiensis*, *Ca. crocodilus* and *Cr. acutus* ($r_s = -0.9394$; $-0.9291$; $-0.9291$ respectively; all $p < 0.0001$; figure 12). Out-lever number negatively correlates with tooth position in *G. gangeticus* ($r_s = -0.54$, $p = 0.0004$) and *V. rudicollis* ($r_s = -0.5046$, $p = 0.0027$) but with weaker relationships (figure 12).

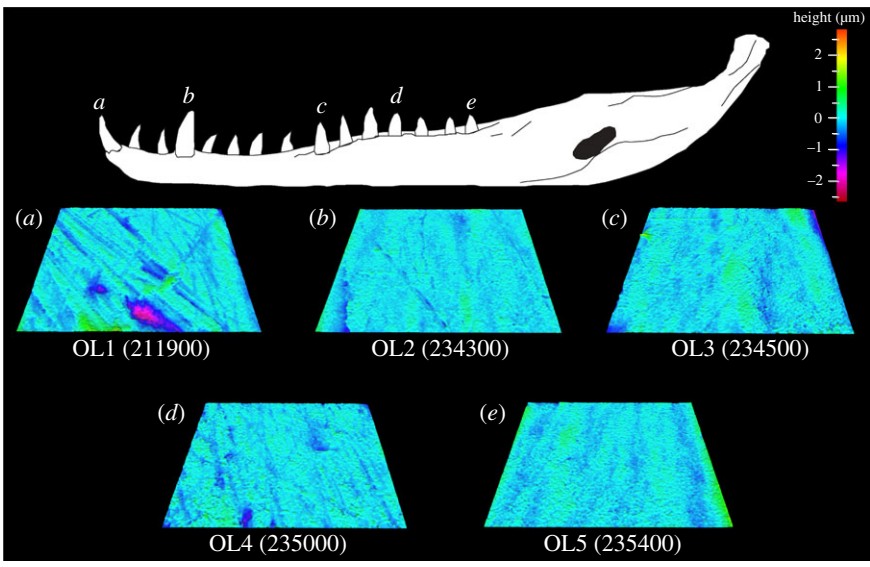

**Figure 6.** Example scale-limited tooth surfaces along the *Crocodylus acutus* tooth row. (*a*) mesial; (*b*) mesial–middle; (*c*) middle; (*d*) middle–distal and (*e*) distal. Example surfaces are illustrative and do not all originate from the same individual. Topographic scale in micrometres. Numbers within brackets denote Leicester IFM sample numbers. The *Cr. acutus* mandible was traced from fig. 1 of [27] under a Creative Commons Attribution 4.0 International License (https://creativecommons.org/licenses/by/4.0/).

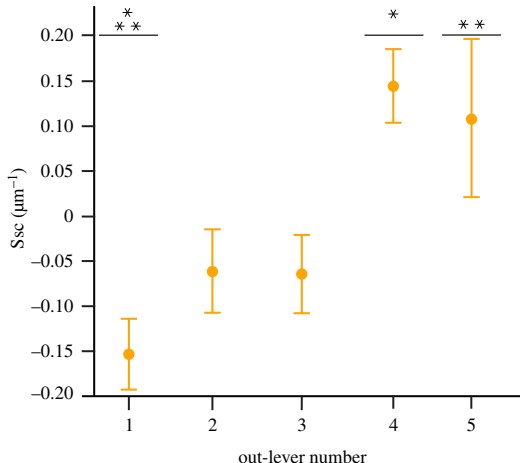

**Figure 7.** Microwear textural differences (mean ± s.e.) along the *Crocodylus acutus* tooth row for each ISO parameter that significantly differs between tooth positions (ANOVA). Out-lever numbers 1–5 denote mesial, mesial–middle, middle, middle–distal and distal tooth positions, respectively. Asterisks represent species-level significant Tukey HSD pairwise differences between out-levers for each parameter.

In *A. mississippiensis,* all ten texture parameters that significantly differ between tooth positions positively correlate with the out-lever number and mechanical advantage, and negatively correlate with aspect ratio ($p < 0.05$; table 7). In *Ca. crocodilus,* all eight parameters that differ between tooth positions positively correlate with out-lever number and negatively correlate with aspect ratio ($p < 0.05$; table 8). Six parameters positively correlate with mechanical advantage ($p < 0.05$; table 8), while Sdq and Sdr do not ($p > 0.05$). In *Cr. acutus,* Ssc positively correlates with out-lever number ($r_s = 0.688$, $p < 0.0001$) and mechanical advantage ($r_s = 0.6496$, $p = 0.0002$), and negatively correlates with aspect ratio ($r_s = -0.6252$, $p = 0.0005$; table 9). In *G. gangeticus,* none of the five parameters that differ between tooth positions correlate with out-lever number, mechanical advantage or aspect ratio ($p > 0.05$; table 10). In *V. rudicollis,* all eight parameters that differ between tooth positions negatively correlate with out-lever number ($p < 0.05$; table 11). Seven parameters positively correlate with mechanical advantage ($p < 0.05$; table 11), while S5z does not. No parameters correlate with aspect ratio ($p > 0.05$; table 11).

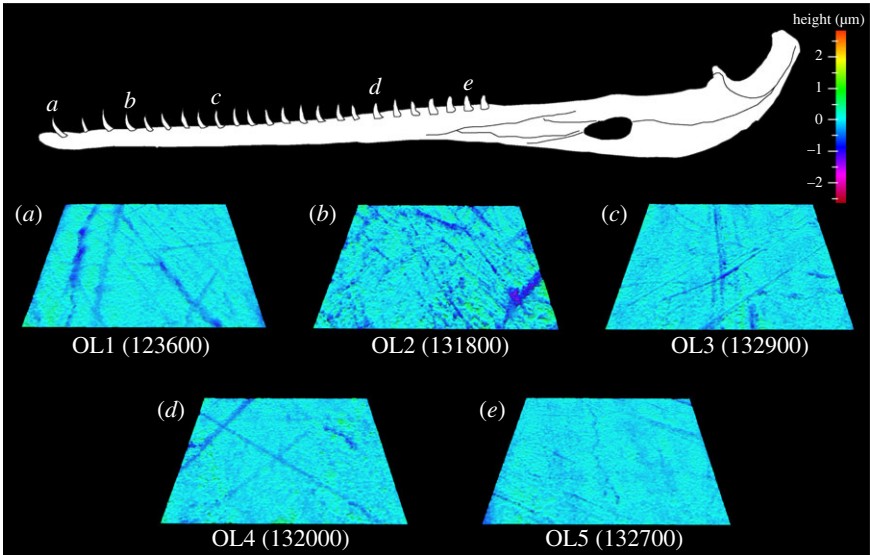

**Figure 8.** Example scale-limited tooth surfaces along the *Gavialis gangeticus* tooth row. (*a*) mesial; (*b*) mesial–middle; (*c*) middle; (*d*) middle–distal and (*e*) distal. Example surfaces are illustrative and do not all originate from the same individual. Topographic scale in micrometres. Numbers within brackets denote Leicester IFM sample numbers. The *G. gangeticus* mandible was traced from fig. 1 of [27] under a Creative Commons Attribution 4.0 International License (https://creativecommons.org/licenses/by/4.0/).

## 3.3. Out-lever texture–dietary space projections

Overall, the projected OL2–5 texture data plot within the bounds of the extant reptile OL1 data (figures 13–17). The only exceptions are with *Ca. crocodilus*; one OL4 tooth plots just beyond the reptiles along PC axis 1 and one OL3 tooth plots just beyond along PC 2 (figure 14). For *A. mississippiensis*, PC 1 differs between out-levers ($F = 5.9443$, d.f. = 4, 34, $p = 0.001$) with OL4 exhibiting more positive values than OLs 1 and 2 (figure 13). PC 2 does not differ between out-levers ($F = 1.2391$, d.f. = 4, 34, $p = 0.3129$). For *Ca. crocodilus*, PCs 1 and 2 do not differ between out-levers (PC 1, $F = 1.0725$, d.f. = 4, 31, $p = 0.3869$; PC 2, $F = 0.2822$, d.f. = 4, 31, $p = 0.8873$; figure 14). For *Cr. acutus*, PCs 1 and 2 do not differ between out-levers (PC 1, $F = 1.319$, d.f. = 4, 22, $p = 0.294$; PC 2, $F = 1.6121$, d.f. = 4, 22, $p = 0.2066$; figure 15). For *G. gangeticus*, PC 1 does not differ between out-levers ($F = 1.1136$, d.f. = 4, 34, $p = 0.3662$). PC 2 differs between out-levers (Welch's ANOVA; $F = 4.3728$, d.f. = 4, 15.78, $p = 0.0143$), with OL3 exhibiting more positive values than OL1 (figure 16). For *V. rudicollis*, PCs 1 and 2 do not differ between out-levers (PC 1, $F = 2.2707$, d.f. = 4, 28, $p = 0.0867$; PC 2, $F = 1.4903$, d.f. = 4, 28, $p = 0.2319$; figure 17).

# 4. Discussion

This study provides the first evidence that reptile tooth microwear textures from non-occlusal surfaces differ along the tooth rows of some species, and that texture differences within each species vary differently with tooth position, mechanical advantage and tooth aspect ratio. Causal mechanisms of non-dietary variables on microwear formation have focused on the occlusal facets of mammalian teeth [8] and are not fully understood [6,47]. However, the results of this study provide the basis for discussions on the relationships between microwear textures and the dietary and non-dietary variables that may affect its development. The findings of this study can therefore be discussed in the contexts of previous DMTA studies, reptile dietary analyses and biomechanical models to (i) understand the influence of non-dietary variables on intraspecific microwear differences within tooth rows, (ii) assess the need for standardized sampling positions for taxa with non-occlusal dentitions, and (iii) identify texture parameters that are more or less useful for dietary reconstructions.

## 4.1. The roles of non-dietary variables on microwear texture differences along reptile tooth rows

In broad terms, there are no discernible patterns regarding the diet, degree of heterodonty or behavioural acquisition of foodstuffs between the reptile species that exhibit microwear texture differences along

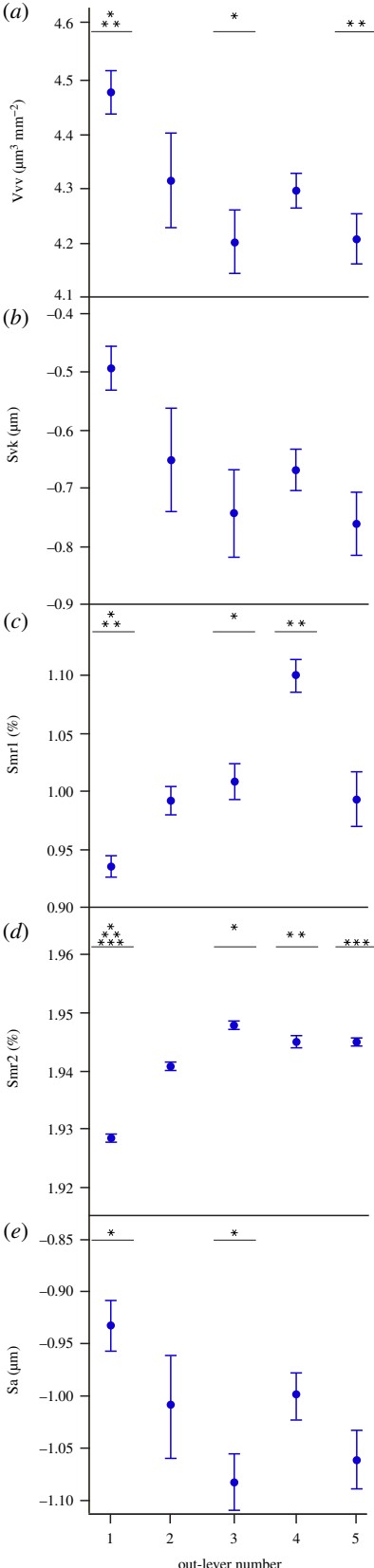

**Figure 9.** Microwear textural differences (mean ± s.e.) along the *Gavialis gangeticus* tooth row for each ISO parameter that significantly differs between tooth positions (ANOVA). Out-lever numbers 1–5 denote mesial, mesial–middle, middle, middle–distal and distal tooth positions, respectively. Asterisks represent species-level significant Tukey HSD pairwise differences between out-levers for each parameter. N.B. The parameter Svk exhibits a significant difference between out-levers (Welch ANOVA) but no significant pairwise differences.

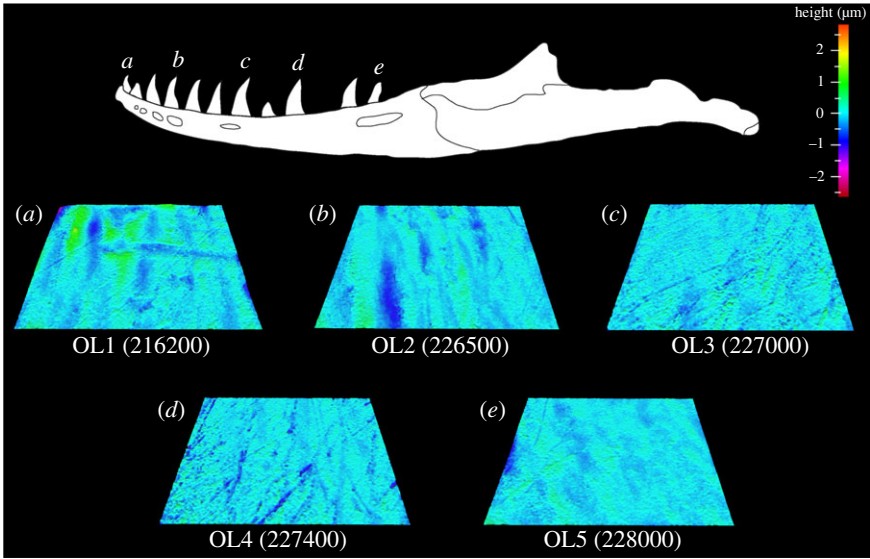

**Figure 10.** Example scale-limited tooth surfaces along the *Varanus rudicollis* tooth row. (*a*) mesial; (*b*) mesial–middle; (*c*) middle; (*d*) middle–distal and (*e*) distal. Example surfaces are illustrative and do not all originate from the same individual. Topographic scale in micrometres. Numbers within brackets denote Leicester IFM sample numbers. The *V. rudicollis* mandible was traced from fig. 1 of [27] under a Creative Commons Attribution 4.0 International License (https://creativecommons.org/licenses/by/4.0/).

tooth rows and the species that do not. Both groups include piscivores (e.g. *G. gangeticus* and *Cr. niloticus* respectively [71,72]), generalist predators (e.g. *V. rudicollis* and *V. salvator* [73,74]) and 'harder' invertebrate consumers (e.g. *Cr. acutus* and *V. olivaceus* [75,76]). Both groups include taxa that exhibit high degrees of tooth heterodonty (e.g. *A. mississippiensis* and *Cr. porosus* adults [40]), and low degrees of heterodonty (e.g. *V. rudicollis* and *V. salvator* [56]). Lastly, crocodilians generally use one side of their jaws to acquire food items, i.e. unilateral biting motions (no preferences are given for using the left or right sides of their jaws) [45], whereas varanids use both sides at once, i.e. bilateral motions [77,78]. This lack of distinguishing factors between taxa that do and do not exhibit texture differences indicates complex variation in microwear differences that is not constrained by phylogenetic relatedness or ecological similarity [79,80]. At present, our analyses do not enable robust interpretations on why no texture differences were found in the eight reptile species as the endogenous and exogenous mechanisms that underlie tooth texture formation remain poorly understood. Further investigation is warranted into whether these findings represent true negatives or are the result of low sample sizes. The rest of the discussion therefore focuses on the five species that did exhibit texture differences along tooth rows.

Microwear texture differences, and the strong correlations of these differences with non-dietary variables, in several crocodilians are somewhat consistent with observational feeding studies and biomechanical models. The middle–distal and distal teeth of *A. mississippiensis* and *Ca. crocodilus*, for example, exhibit the roughest microwear textures with textural differences positively correlating with mechanical advantage and negatively correlating with aspect ratio. Of the ten and eight parameters that varied along the *A. mississippiensis* and *Ca. crocodilus* tooth rows, respectively, only two from the former (Spk and Vmp) and none from the latter differed between the extant reptile dietary guilds of [10]. This provides some evidence of intra-jaw dietary partitioning in *A. mississippiensis* and no partitioning in *Ca. crocodilus*, and that microwear texture differences are probably primarily due to non-dietary factors. Many crocodilians that exhibit higher degrees of tooth heterodonty, including alligators and caimans [40], exhibit preferential tooth use when handling food items [51,81–83]. The more conical and slender teeth in the mesial regions of the tooth row are primarily used to seize and move food items towards the throat, and the shorter, more bulbous teeth in the distal regions are used to crush larger and/or 'harder' items before swallowing [45,51,84]. Distally positioned teeth in these species can therefore be expected to experience slightly more frequent tooth–food interactions. Distally positioned teeth also experience higher bite forces during jaw closure [45,57,85]. Comparable trends are reported in New World monkeys (Platyrrhini) where the non-occlusal surface textures of canines, which are preferentially used to process fruits and experience higher bite forces, exhibit rougher textures than the incisors [32]. That our results are consistent with these studies suggest that DMTA is tracking behavioural differences in tooth use along tooth rows, as opposed to dietary differences *per se*.

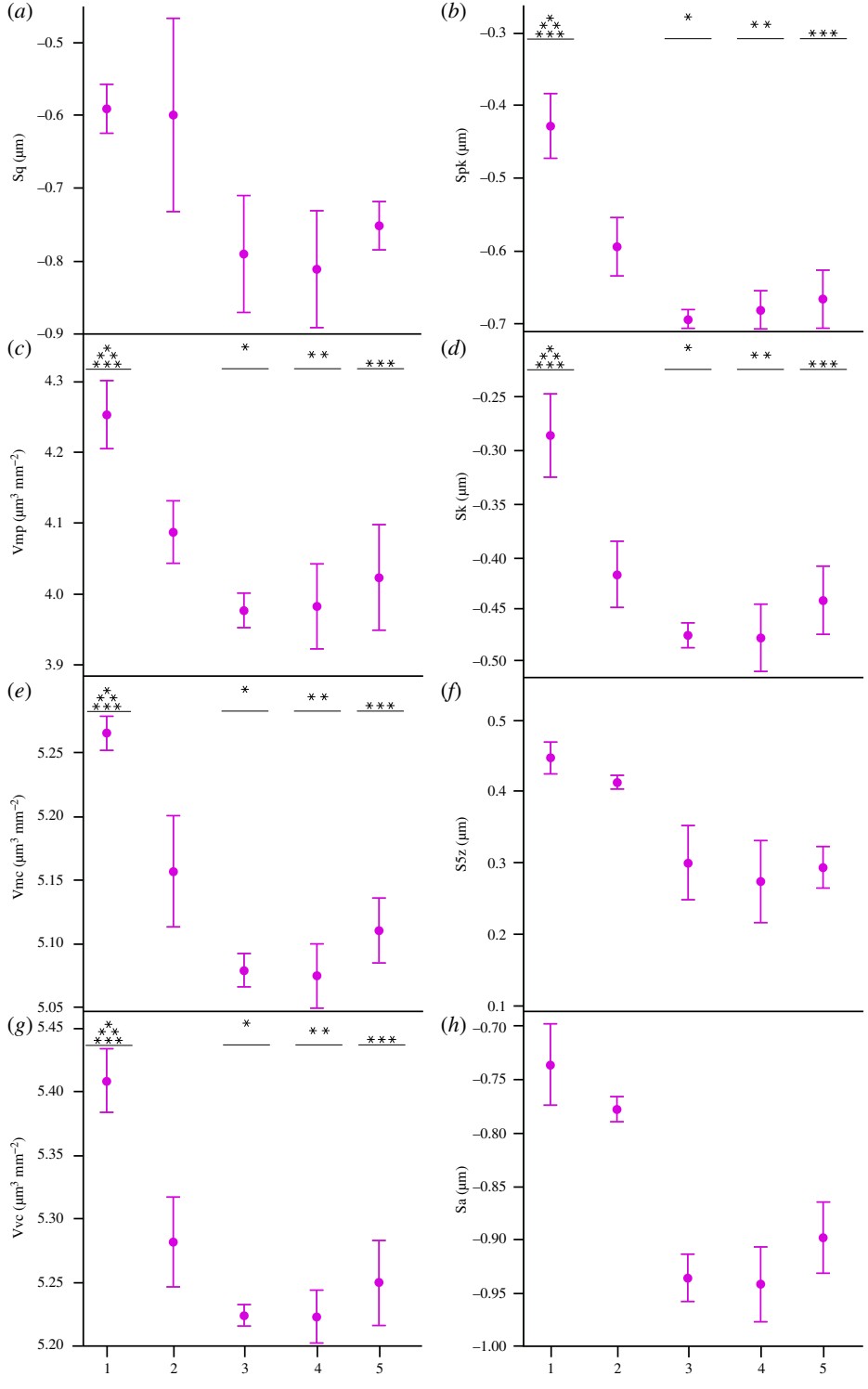

**Figure 11.** Microwear textural differences (mean ± s.e.) along the *Varanus rudicollis* tooth row for each ISO parameter that significantly differs between tooth positions (ANOVA). Out-lever numbers 1–5 denote mesial, mesial–middle, middle, middle–distal and distal tooth positions, respectively. Asterisks represent species-level significant Tukey HSD pairwise differences between out-levers for each parameter. N.B. Parameters Sq, S5z and Sa exhibit significant differences between out-levers (Welch ANOVA) but no significant pairwise differences.

A similar conclusion can be drawn for *Cr. acutus*, albeit with less certainty, since only one texture parameter, Ssc, differs along with its tooth row. *Crocodylus acutus* consume the highest proportions of 'harder' invertebrates, primarily crabs [76,86], of all the study reptiles. High bite forces are required to fracture invertebrate exoskeletons [87–89], thus preferential tooth use in seizing and/or crushing items

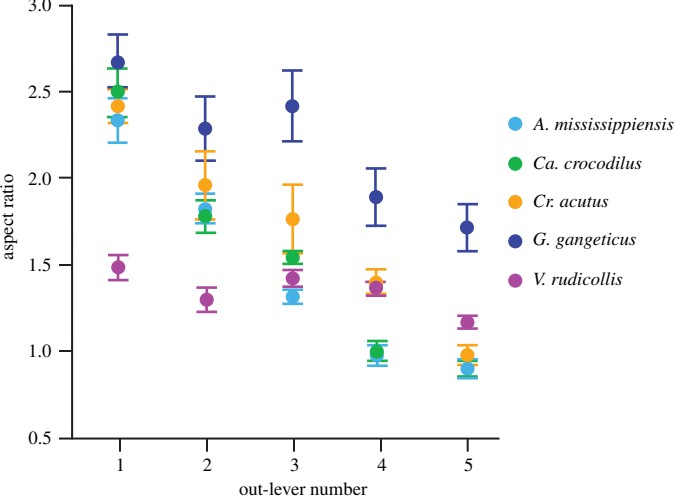

**Figure 12.** Mean tooth aspect ratios (±s.e.) between tooth positions in the five reptile species that exhibit microwear texture differences along the tooth row. Tooth positions are denoted by out-lever number where out-levers 1–5 denote mesial, mesial–middle, middle, middle–distal and distal teeth, respectively.

would probably have resulted in vastly different microwear textures along tooth rows. Why the distally positioned teeth only exhibited significantly rounder textural peaks than the mesial teeth remains unclear. Furthermore, Ssc significantly differed in only one of the five species that exhibited textural differences. This indicates complex variation in microwear textures and non-dietary variables that should be considered in future studies.

In contrast with these crocodilians, the mesial and mesial-most teeth of *V. rudicollis* exhibit the roughest microwear textures. Of the eight parameters that differed along the tooth row, two parameters (Spk and Vmp) differed between the dietary guilds of [10] and four (Sa, Spk, Sq, Vmc and Vvc) differed between the guilds of [11]. This provides potential evidence of intra-jaw dietary partitioning in this reptile. However, possible influences of non-dietary variables and behavioural tooth usage cannot be ruled out. For example, *V. rudicollis* predate upon taxa much smaller than themselves, such as frogs (Anura) and grasshoppers (Orthoptera) [73]. Captured food items are thus usually swallowed whole with minimal oral handling or processing [90]. This, coupled with bilateral biting motions typical of most varanids [77,78], result in the mesially positioned teeth experiencing the most frequent tooth–food interactions. These teeth are also subjected to the lowest bite forces during jaw closure [42,50,53]. This, combined with preferential tooth use, could explain why seven parameters correlate with mechanical advantage. That none of the eight parameters correlated with tooth aspect ratio is not unexpected as *V. rudicollis* exhibits the lowest degree of tooth heterodonty of the reptiles that exhibited tooth row textural differences. However, rougher distal tooth textures may have been expected due to mechanical differences. The relatively slender distal teeth of *V. rudicollis* are subjected to proportionally higher loading pressures than the rounder, bulbous distal teeth of other species (e.g. alligators and caimans) during jaw closure due to smaller tooth–food contact areas [42,50,53]. Our results indicate that *V. rudicollis* texture differences were not influenced by less pronounced tooth shape differences and further highlights the complex variation in microwear textures and non-dietary variables between species.

This complexity is further highlighted in *G. gangeticus* as the mesially positioned teeth exhibited the highest volume and height parameter values, while the middle teeth (and, to a lesser extent, the distal teeth) exhibited the highest material ratio values. No single region(s) of the *G. gangeticus* tooth row can therefore be argued to have the roughest textures. Of the five parameters that exhibited tooth row differences, only one (Smr1) differed between the dietary guilds of [10], and two (Sa and Vvv) differed between the guilds of [11]. The multivariate framework of the latter, however, does not include crocodilians and so is not as representative a comparison. Nevertheless, these similarities indicate some level of intra-jaw dietary partitioning. *Gavialis gangeticus* is widely regarded as an almost exclusive piscivore [71], although tetrapods are noted to be consumed on occasion [91]. Biomechanical models show that the elongate *G. gangeticus* rostrum is subjected to much higher feeding-related stresses and strains than other crocodilians [92,93]. It is therefore tempting to suggest that fish and occasional tetrapods are captured using different teeth to reduce mechanical loads on *G. gangeticus* skulls (see Out-

**Table 7.** Spearman's rank correlations (4 d.p.) between ISO texture parameters that significantly differ between tooth positions of the *Alligator mississippiensis* tooth row and the non-dietary variables; tooth aspect ratio, out-lever number and mechanical advantage. Correlations exhibiting significant differences after the Benjamini–Hochberg procedure shown in italics.

| parameter | non-dietary variable | $\rho$ value | $p$ value |
| --- | --- | --- | --- |
| Sq | aspect ratio | −0.4479 | *0.0042* |
| | out-lever number | 0.4442 | *0.0046* |
| | mechanical advantage | 0.4916 | *0.0015* |
| Sp | aspect ratio | −0.5718 | *0.0001* |
| | out-lever number | 0.5335 | *0.0005* |
| | mechanical advantage | 0.508 | *0.001* |
| Sz | aspect ratio | −0.4628 | *0.003* |
| | out-lever number | 0.4889 | *0.0016* |
| | mechanical advantage | 0.5457 | *0.0003* |
| Vmp | aspect ratio | −0.5557 | *0.0002* |
| | out-lever number | 0.4793 | *0.002* |
| | mechanical advantage | 0.4492 | *0.0041* |
| Vmc | aspect ratio | −0.4184 | *0.008* |
| | out-lever number | 0.4605 | *0.0032* |
| | mechanical advantage | 0.5605 | *0.0002* |
| Vvc | aspect ratio | −0.4526 | *0.0038* |
| | out-lever number | 0.4468 | *0.0043* |
| | mechanical advantage | 0.5012 | *0.0012* |
| Spk | aspect ratio | −0.5561 | *0.0002* |
| | out-lever number | 0.4889 | *0.0016* |
| | mechanical advantage | 0.464 | *0.0029* |
| Sk | aspect ratio | −0.3864 | *0.0151* |
| | out-lever number | 0.4429 | *0.0047* |
| | mechanical advantage | 0.534 | *0.0005* |
| S5z | aspect ratio | −0.5599 | *0.0002* |
| | out-lever number | 0.5851 | *<0.0001* |
| | mechanical advantage | 0.6324 | *<0.0001* |
| Sa | aspect ratio | −0.434 | *0.0058* |
| | out-lever number | 0.4304 | *0.0062* |
| | mechanical advantage | 0.4992 | *0.0012* |

lever dietary projections). The lack of influence of non-dietary variables on microwear texture difference also lends support to this suggestion. *Gavialis gangeticus* exhibits the lowest degree of tooth heterodonty of all extant crocodilians [40], and tooth pressure experiments show comparatively uniform bite forces along the tooth row [45]. The mechanical forces experienced by *G. gangeticus* could therefore be argued to be relatively similar across the tooth row and have less impact on microwear formation, as our DMTA results suggest. This is not a novel suggestion as intraspecific microwear texture differences in xenarthran mammals (Xenarthra) also did not correlate with tooth position and mechanical advantage [29–31]. However, the influence of specialized feeding behaviours on microwear formation cannot be ruled out. *Gavialis gangeticus* captures fish by sweeping its elongate jaws in rapid lateral strikes [94]. Its jaws are therefore better adapted for rapid closure than for shaking or crushing food items, as in other crocodilians [92,93,95]. While the impact of these behaviours on microwear formation is unknown, the results of this study further highlight the potential for DMTA to track behavioural differences in tooth use and the need for thorough appreciations of the study species.

**Table 8.** Spearman's rank correlations (4 d.p.) between ISO texture parameters that significantly differ between tooth positions of the *Caiman crocodilus* tooth row and the non-dietary variables; tooth aspect ratio, out-lever position and mechanical advantage. Correlations exhibiting significant differences after the Benjamini–Hochberg procedure shown in italics.

| parameter | non-dietary variable | ρ value | p value |
|---|---|---|---|
| Sq | aspect ratio | −0.474 | *0.0035* |
| | out-lever number | 0.447 | *0.0063* |
| | mechanical advantage | 0.4013 | *0.0153* |
| Sdq | aspect ratio | −0.3526 | *0.0326* |
| | out-lever number | 0.3546 | *0.0338* |
| | mechanical advantage | 0.3095 | 0.0663 |
| Sdr | aspect ratio | −0.3526 | *0.0349* |
| | out-lever number | 0.3515 | *0.0356* |
| | mechanical advantage | 0.3071 | 0.0685 |
| Vmc | aspect ratio | −0.3955 | *0.017* |
| | out-lever number | 0.4173 | *0.0113* |
| | mechanical advantage | 0.3558 | *0.0332* |
| Vvc | aspect ratio | −0.4399 | *0.0073* |
| | out-lever number | 0.4348 | *0.0081* |
| | mechanical advantage | 0.3719 | *0.0255* |
| Sk | aspect ratio | −0.3919 | *0.0166* |
| | out-lever number | 0.415 | *0.0118* |
| | mechanical advantage | 0.3586 | *0.0317* |
| Svk | aspect ratio | −0.3854 | *0.0203* |
| | out-lever number | 0.3932 | *0.0177* |
| | mechanical advantage | 0.3554 | *0.0334* |
| Sa | aspect ratio | −0.4453 | *0.0065* |
| | out-lever number | 0.4411 | *0.0071* |
| | mechanical advantage | 0.3855 | *0.0202* |

**Table 9.** Spearman's rank correlations (4 d.p.) between ISO texture parameters that significantly differ between tooth positions of the *Crocodylus acutus* tooth row and the non-dietary variables; tooth aspect ratio, out-lever position and mechanical advantage. Correlations exhibiting significant differences after the Benjamini–Hochberg procedure shown in italics.

| parameter | non-dietary variable | ρ value | p value |
|---|---|---|---|
| Ssc | aspect ratio | −0.6252 | *0.0005* |
| | out-lever number | 0.688 | *<0.0001* |
| | mechanical advantage | 0.6496 | *0.0002* |

## 4.2. Out-lever dietary projections

The out-lever projections into the texture–dietary space further highlight complex variation along the tooth rows of different reptile species, which has important implications for standardized sampling positions. In *A. mississippiensis*, for example, the separation of out-levers along PC 1 (figure 13) provides additional evidence for some level of intra-jaw dietary partitioning. The middle–distal teeth interact with significantly more invertebrates, probably 'harder' invertebrates, than the mesial and mesial–middle teeth. These results corroborate previous observations and suggestions of such preferential tooth use in *A. mississippiensis* [51,81–83] and highlight how the location of sampled teeth can lead to different dietary reconstructions.

**Table 10.** Spearman's rank correlations (4 d.p.) between ISO texture parameters that significantly differ between tooth positions of the *Gavialis gangeticus* tooth row and the non-dietary variables; tooth aspect ratio, out-lever position and mechanical advantage. None of the correlations exhibited significant differences after the Benjamini–Hochberg procedure.

| parameter | non-dietary variable | $\rho$ value | *p* value |
|---|---|---|---|
| Vvv | aspect ratio | 0.0481 | 0.7713 |
| | out-lever number | −0.4115 | 0.0093 |
| | mechanical advantage | −0.2929 | 0.0703 |
| Svk | aspect ratio | 0.0357 | 0.829 |
| | out-lever number | −0.3906 | 0.014 |
| | mechanical advantage | −0.2965 | 0.0668 |
| Smr1 | aspect ratio | −0.1587 | 0.3346 |
| | out-lever number | 0.3884 | 0.0145 |
| | mechanical advantage | 0.104 | 0.5287 |
| Smr2 | aspect ratio | −0.0737 | 0.6557 |
| | out-lever number | 0.3975 | 0.0122 |
| | mechanical advantage | 0.1743 | 0.2887 |
| Sa | aspect ratio | −0.0133 | 0.9359 |
| | out-lever number | −0.2906 | 0.0727 |
| | mechanical advantage | −0.2688 | 0.098 |

*Gavialis gangeticus* also shows, to a lesser extent, evidence of intra-jaw dietary partitioning. The separation of out-levers along PC 2 (figure 16) indicates that the middle teeth interact with 'softer' invertebrates than the mesial teeth (based on the correlations between the PC axes and diet [10]). While some unbeknownst dietary diversity cannot be ruled out, that the middle teeth also exhibit greater degrees of overlap with the carnivore guild in the texture–dietary space indicates that these teeth may be preferentially used to handle the occasional tetrapod [71,91]. This is somewhat surprising, however, as the distal areas of the tooth row are the most resistant against the stresses and strains associated with handling larger food items [92,93]. Why the middle–distal and distal tooth textures do not exhibit any dietary signals is unclear, but these results nevertheless show the importance of tooth sampling location for representative analyses, even for taxa that exhibit high levels of ecological specialism.

In contrast with these crocodilians, the *V. rudicollis* out-lever projections into the texture–dietary space cast doubt on intra-jaw dietary partitioning in this species (figure 17). The parameters Spk and Vmp differ along both the *V. rudicollis* and *A. mississippiensis* tooth rows and between the dietary guilds of [10]. However, the lack of separation of *V. rudicollis* out-levers along both PC axes (figure 17) indicates that the intraspecific texture differences in the monitor lizard are unlikely to show a dietary signal. Mechanical and/or behavioural signals are therefore more likely preserved in *V. rudicollis* tooth rows. While it could be argued that standardized sampling positions are not needed for this species, future dietary analyses that include multiple taxa should nevertheless account for these non-dietary signals for more representative reconstructions.

The lack of separation between the *Ca. crocodilus* and *Cr. acutus* out-levers within the texture–dietary space (figures 14 and 15, respectively) is expected since none of the parameters that exhibited tooth row differences in these species differed between the dietary guilds of [10]. Standardized sampling positions could therefore be argued as less necessary for these species.

Overall, these results show that sampling teeth from different positions of reptile tooth rows has the potential to influence dietary reconstructions, sometimes in unintuitive ways, and even mask dietary signals in microwear textures. It could therefore be argued that standardized sampling positions are best adopted for reptiles and, by extension, taxa with non-occlusal dentitions. Isolated teeth of extant and extinct taxa, especially species that exhibit low degrees of tooth heterodonty, should therefore only be sampled if their original locations in the tooth row are known. Robust sampling strategies will need thorough appreciations for the study taxon or taxa, and for the research question(s) being asked [34,36]. Dietary reconstructions, for example, could focus on teeth that are less affected by non-dietary

**Table 11.** Spearman's rank correlations (4 d.p.) between ISO texture parameters that significantly differ between tooth positions of the *Varanus rudicollis* tooth row and the non-dietary variables; tooth aspect ratio, out-lever position and mechanical advantage. Correlations exhibiting significant differences after the Benjamini–Hochberg procedure shown in italics.

| parameter | non-dietary variable | $\rho$ value | $p$ value |
| --- | --- | --- | --- |
| Sq | aspect ratio | 0.1998 | 0.265 |
| | out-lever number | −0.5273 | *0.0016* |
| | mechanical advantage | −0.4132 | *0.0169* |
| Vmp | aspect ratio | 0.3573 | 0.0412 |
| | out-lever number | −0.5204 | *0.0019* |
| | mechanical advantage | −0.4439 | *0.0097* |
| Vmc | aspect ratio | 0.1686 | 0.3482 |
| | out-lever number | −0.584 | *0.0016* |
| | mechanical advantage | −0.3864 | *0.0264* |
| Vvc | aspect ratio | 0.2201 | 0.2184 |
| | out-lever number | −0.4923 | *0.0036* |
| | mechanical advantage | −0.3546 | *0.0429* |
| Spk | aspect ratio | 0.3665 | 0.0359 |
| | out-lever number | −0.5274 | *0.0016* |
| | mechanical advantage | −0.4401 | *0.0104* |
| Sk | aspect ratio | 0.1802 | 0.3157 |
| | out-lever number | −0.539 | *0.0012* |
| | mechanical advantage | −0.3981 | *0.0218* |
| S5z | aspect ratio | 0.3052 | 0.0841 |
| | out-lever number | −0.4438 | *0.0097* |
| | mechanical advantage | −0.3272 | 0.0631 |
| Sa | aspect ratio | 0.1515 | 0.4 |
| | out-lever number | −0.5007 | *0.003* |
| | mechanical advantage | −0.3874 | *0.0259* |

variables, and investigations into mechanical and behavioural differences in feeding could focus on teeth that are maximally affected by non-dietary variables. Furthermore, these results show that DMTA has the potential to reconstruct feeding behaviours of extinct taxa, although thorough understandings of the diets and feeding behaviours of extant animals to use as multivariate frameworks will be needed for representative analyses.

## 4.3. International Organization for Standardization texture parameter usefulness in dietary reconstructions

It should be made clear that previous dietary reconstructions of taxa with non-occlusal dentitions are not flawed since standardized sampling positions were used [11,32,33], although little consideration was given to the potential confounding effects of other non-dietary variables besides tooth position. Nevertheless, comparisons between the texture parameters that differ along reptile tooth rows with those that differ along the tooth rows of non-reptilian taxa enable identification of parameters that may be more or less useful for dietary reconstructions.

Of the 17 ISO parameters that differed in at least one reptile tooth row in this study, five parameters differed in three or more species (Sa, Sk, Sq, Vmc and Vvc). These parameters could therefore be argued to be more strongly influenced by the non-dietary variables investigated in this study. Conversely, Sds, Sku, Str and Sv do not differ along the tooth rows of any species and could thus be argued to be less

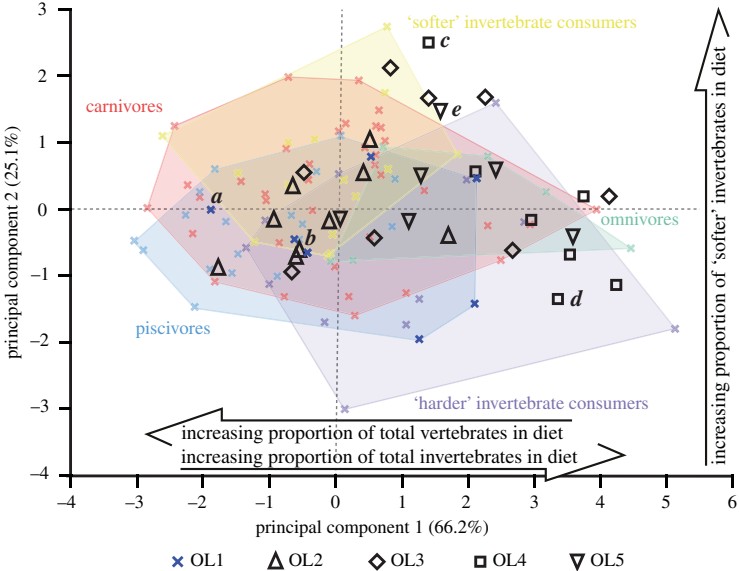

**Figure 13.** Texture–dietary space of International Organization for Standardization texture parameters for reptile dietary guilds and *Alligator mississippiensis* out-levers. Texture–dietary space based on reptile data with *A. mississippiensis* out-levers 2–5 projected onto the first two axes as unknown datum points. OL1 data form part of the piscivore guild and are highlighted. Datum points labelled *a*–*e* correspond to the scale-limited tooth textures shown in figure 2. Arrows show significant correlations of dietary characteristics along PC axes 1 and 2. Texture–dietary space adapted from fig. 2 of [10] under a Creative Commons Attribution 4.0 International License https://creativecommons.org/licenses/by/4.0/ to include the OL2–5 data.

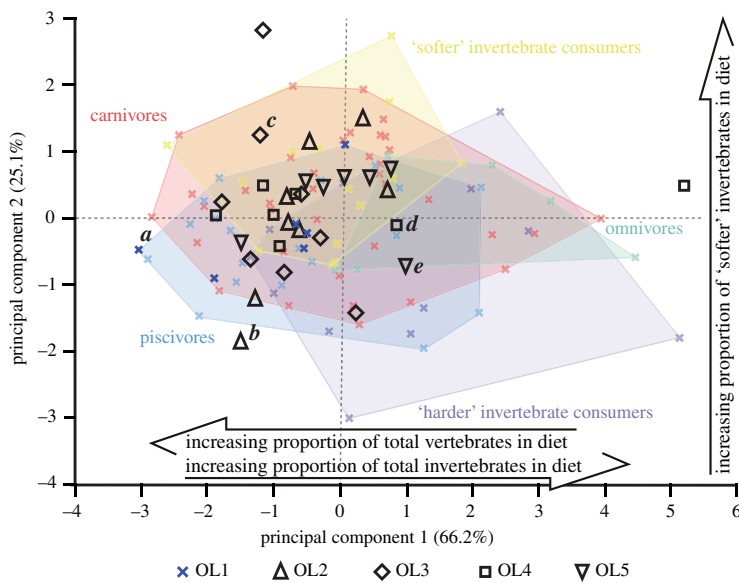

**Figure 14.** Texture–dietary space of International Organization for Standardization texture parameters for reptile dietary guilds and *Caiman crocodilus* out-levers. Texture–dietary space based on reptile data with *Ca. crocodilus* out-levers 2–5 projected onto the first two axes as unknown datum points. OL1 data form part of the piscivore guild and are highlighted. Datum points labelled *a*–*e* correspond to the scale-limited tooth textures shown in figure 4. Arrows show significant correlations of dietary characteristics along PC axes 1 and 2. Texture–dietary space adapted from fig. 2 of [10] under a Creative Commons Attribution 4.0 International License https://creativecommons.org/licenses/by/4.0/ to include the OL2–5 data.

strongly influenced by non-dietary variables. More broadly, DMTA of beluga whale (*Delphinapterus leucas*) teeth found significant pairwise differences in Sa, Sds, Sk, Sku, Sq, Vmc and Vvc along with tooth rows [34]. Four of these parameters (Sa, Sk, Sq and Vmc) also differ in at least two reptiles from the present study. Similarly, DMTA of killer whale (*Orcinus orca*) teeth found that Sa, Sk, Sq, Vmc,

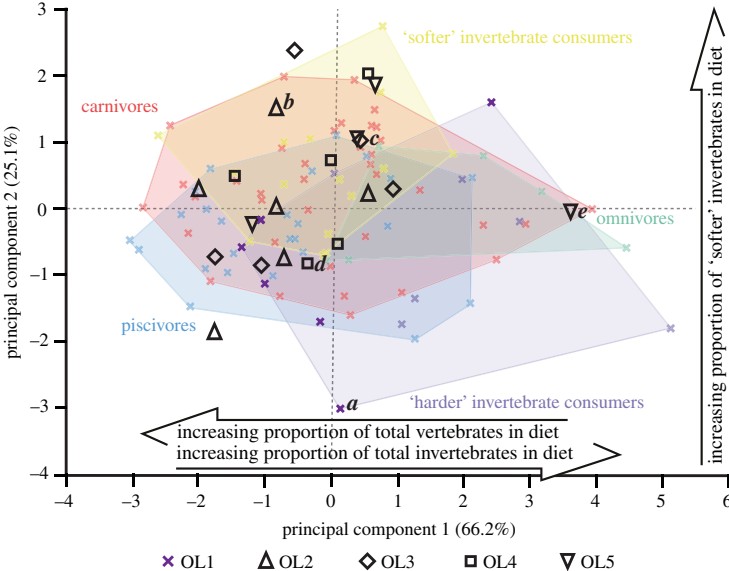

**Figure 15.** Texture–dietary space of International Organization for Standardization texture parameters for reptile dietary guilds and *Crocodylus acutus* out-levers. Texture–dietary space based on reptile data with *Cr. acutus* out-levers 2–5 projected onto the first two axes as unknown datum points. OL1 data form part of the 'harder' invertebrate consumer guild and are highlighted. Datum points labelled *a–e* correspond to the scale-limited tooth textures shown in figure 6. Arrows show significant correlations of dietary characteristics along PC axes 1 and 2. Texture–dietary space adapted from fig. 2 of [10] under a Creative Commons Attribution 4.0 International License https://creativecommons.org/licenses/by/4.0/ to include the OL2–5 data.

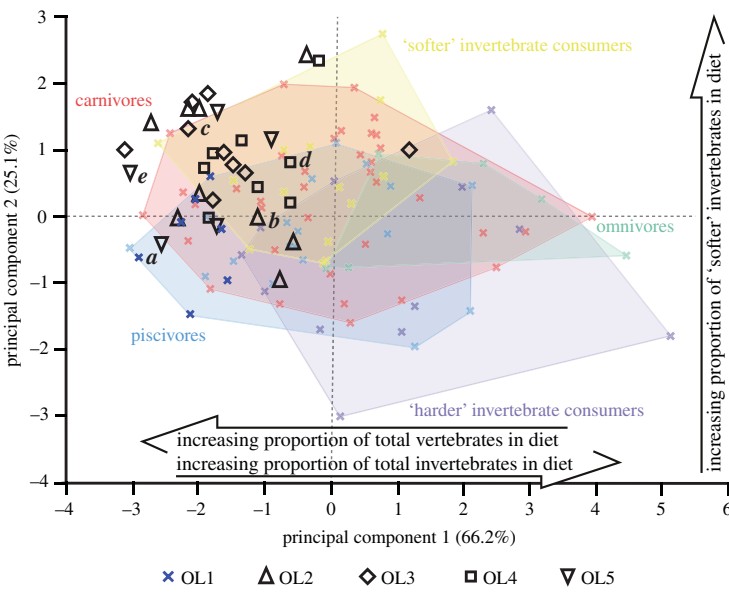

**Figure 16.** Texture–dietary space of International Organization for Standardization texture parameters for reptile dietary guilds and *Gavialis gangeticus* out-levers. Texture–dietary space based on reptile data with *G. gangeticus* out-levers 2–5 projected onto the first two axes as unknown datum points. OL1 data form part of the piscivore guild and are highlighted. Datum points labelled *a–e* correspond to the scale-limited tooth textures shown in figure 8. Arrows show significant correlations of dietary characteristics along PC axes 1 and 2. Texture–dietary space adapted from fig. 2 of [10] under a Creative Commons Attribution 4.0 International License https://creativecommons.org/licenses/by/4.0/ to include the OL2–5 data.

Vvc and Vvv exhibited the greatest differences between tooth positions (largest effect sizes in ANOVA and greatest numbers of pairwise differences) [33]. This partial overlap of parameters between reptiles and odontocetes provide further evidence that Sa, Sk, Sq and Vmc are perhaps the most likely to be influenced by non-dietary variables. This study is not saying that these parameters should not be

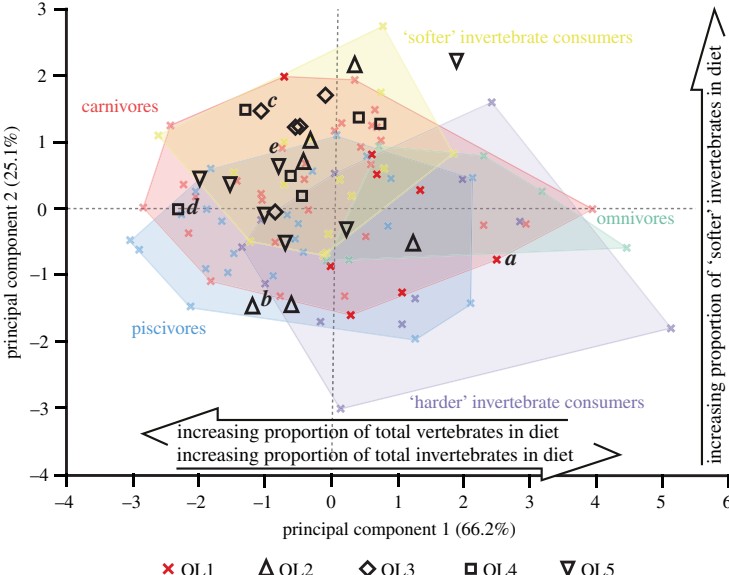

**Figure 17.** Texture–dietary space of International Organization for Standardization texture parameters for reptile dietary guilds and *Varanus rudicollis* out-levers. Texture–dietary space based on reptile data with *V. rudicollis* out-levers 2–5 projected onto the first two axes as unknown datum points. OL1 data form part of the carnivore guild and are highlighted. Datum points labelled *a–e* correspond to the scale-limited tooth textures shown in figure 10. Arrows show significant correlations of dietary characteristics along PC axes 1 and 2. Texture–dietary space adapted from fig. 2 of [10] under a Creative Commons Attribution 4.0 International License https://creativecommons.org/licenses/by/4.0/ to include the OL2–5 data.

used in future dietary reconstructions, but rather that any significant differences exhibited by these parameters should be interpreted with caution and with appreciations for the study species.

## 5. Conclusion

This study demonstrates that reptile tooth microwear textures from non-occlusal surfaces significantly differ along the tooth rows of some species and the complex influence of endogenous non-dietary variables on these texture differences in different species. Some textural differences match those reported along primate, odontocete and xenarthran tooth rows [29–34], while others show trends not previously reported. Non-dietary variables therefore have the potential to obscure dietary signals in some taxa. Consequently, standardizing sampling positions within non-occlusal tooth rows are recommended in order to reduce confounding effects of non-dietary variables in dietary reconstructions with DMTA. This unfortunately limits the applicability of DMTA to isolated teeth of extant and extinct taxa unless there are high levels of confidence in where specimens were originally located in the jaw. Interestingly, this study also provides evidence of mechanical and behavioural signals, with respect to feeding, preserved within tooth microwear textures. By focusing on teeth that are maximally influenced by non-dietary variables, DMTA can potentially also reconstruct the feeding behaviours of extant and extinct taxa in addition to, or instead of, diet. This would provide a more thorough characterization of the ecological roles that taxa perform within ecosystems and more thorough insight into the functioning of modern and ancient ecosystems [10,41,96].

Ethics. All sampled specimens are available in publicly accessible museum collections. The specimen numbers of all studied specimens are available in the electronic supplementary material.
Data accessibility. The raw data generated from this study, including the microwear texture data, jaw in-lever and out-lever measurements and tooth aspect ratio data are available in the electronic supplementary material. No new code was generated from this study.
Authors' contributions. All authors conceived the study and developed the analytical design. J.B. collected the data. J.B. analysed the data with input from D.M.H. and M.A.P. J.B. wrote the paper with contributions from all authors.
Competing interests. The authors declare no competing interests.

Funding. This work was funded by a NERC studentship awarded to J.B. through the Central England NERC Training Alliance (CENTA; grant reference NE/L002493/1) and by the University of Leicester. J.B. was also supported by a Leverhulme Trust Research Project Grant (RPG-2019-364) during the completion and write-up of the project.
Acknowledgements. We thank P. Campbell (NHMUK), M. Carnall (OUMNH), T. Davidson (LDUCZ), C. Sheehy (UF), A. Resetar (FMNH) and A. Wynn (USNM) for access to specimens. Thanks to N. Adams for assistance in producing the DEMs. Thanks to T. Harvey and P. Cox, and two anonymous reviewers, for helpful comments on an earlier version of this manuscript.

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
