## [Peer Review File · Royal Society Open Science]

Review History

RSOS-201754.R0 (Original submission)

Review form: Reviewer 1 (Brian Beatty)

Is the manuscript scientifically sound in its present form?

Yes

Are the interpretations and conclusions justified by the results?

Yes

Is the language acceptable?

Yes

Do you have any ethical concerns with this paper?

No

Have you any concerns about statistical analyses in this paper?

No

Recommendation?

Accept with minor revision (please list in comments)

Comments to the Author(s)

This is a long-awaited carefully done look at microwear in squamates and crocodylians. The authors were careful not to overreach in their interpretations, which is otherwise a tendency with microwear. I applaud their care in making hypotheses clear and actually tested with good control over variables not being tested. This opens a field of study that is overdue for work, and is a pattern for the lab leader Purnell.

I cannot find any major revisions needed, and only one minor revision appears worth any effort to make. Near lines 161-162 the authors refer to sampling the non-occlusal labial surfaces as close to the apex as possible. This needs two points of clarification: 1) how is a non-occlusal surface area identified/defined? 2) As close to for some broken teeth (even if broken post-mortem) might be very close to the base. Do you have a criterion for rejection of a tooth if the break is too far from the apex?

Other than these two issues, I don't see any need for revisions. I urge the editors to accept this paper, I want to cite it.

Review form: Reviewer 2 (Joshua Cohen)**Is the manuscript scientifically sound in its present form?**

Yes

Are the interpretations and conclusions justified by the results?

Yes

Is the language acceptable?

Yes

Do you have any ethical concerns with this paper?

No

Have you any concerns about statistical analyses in this paper?

Yes

Recommendation?

Accept with minor revision (please list in comments)

Comments to the Author(s)

I have attached my comments to the authors in a separate file (Appendix A).

Decision letter (RSOS-201754.R0)

Dear Dr Bestwick

On behalf of the Editors, we are pleased to inform you that your Manuscript RSOS-201754 "Dental microwear texture analysis along reptile tooth rows: complex variation with non-dietary variables" has been accepted for publication in Royal Society Open Science subject to minor revision in accordance with the referees' reports. Please find the referees' comments along with any feedback from the Editors below my signature.

Please submit your revised manuscript and required files (see below) no later than 7 days from today's (ie 18-Nov-2020) date. Note: the ScholarOne system will 'lock' if submission of the revision is attempted 7 or more days after the deadline. If you do not think you will be able to meet this deadline please contact the editorial office immediately.

on behalf of Dr Emily Lindsey (Associate Editor) and Pete Smith (Subject Editor)
openscience@royalsociety.org

Associate Editor Comments to Author (Dr Emily Lindsey):

Associate Editor: 1

Comments to the Author:

This paper received two quite positive reviews and should be a valuable contribution to the field. Please make the revisions suggested by the reviewers.

Reviewer comments to Author:

Reviewer: 1

Comments to the Author(s)

This is a long-awaited carefully done look at microwear in squamates and crocodylians. The authors were careful not to overreach in their interpretations, which is otherwise a tendency with microwear. I applaud their care in making hypotheses clear and actually tested with good control over variables not being tested. This opens a field of study that is overdue for work, and is a pattern for the lab leader Purnell.

I cannot find any major revisions needed, and only one minor revision appears worth any effort to make. Near lines 161-162 the authors refer to sampling the non-occlusal labial surfaces as close to the apex as possible. This needs two points of clarification: 1) how is a non-occlusal surface area identified/ defined? 2) As close to for some broken teeth (even if broken post-mortem) might be very close to the base. Do you have a criterion for rejection of a tooth if the break is too far from the apex?

Other than these two issues, I don't see any need for revisions. I urge the editors to accept this paper, I want to cite it.

Reviewer: 2

Comments to the Author(s)

I have attached my comments to the authors in a separate file.

===PREPARING YOUR MANUSCRIPT===

===PREPARING YOUR REVISION IN SCHOLARONE===

Author's Response to Decision Letter for (RSOS-201754.R0)

See Appendix B.

Decision letter (RSOS-201754.R1)

Dear Dr Bestwick,

It is a pleasure to accept your manuscript entitled "Dental microwear texture analysis along reptile tooth rows: complex variation with non-dietary variables" in its current form for publication in Royal Society Open Science. The comments of the reviewer(s) who reviewed your manuscript are included at the foot of this letter.

on behalf of Dr Emily Lindsey (Associate Editor) and Pete Smith (Subject Editor)
openscience@royalsociety.org

Associate Editor Comments to Author (Dr Emily Lindsey):

This is a well-conducted and clearly-presented study on a topic that will advance the field and be of value to other researchers. The two reviewers provided recommendations for only minor revisions, and the authors have addressed these satisfactorily.

Follow Royal Society Publishing on Twitter: @RSocPublishing
Follow Royal Society Publishing on Facebook:
<https://www.facebook.com/RoyalSocietyPublishing.FanPage/>

Read Royal Society Publishing's blog:
<https://royalsociety.org/blog/blogsearchpage/?category=Publishing>

Appendix A

The manuscript "Dental microwear texture analysis along reptile tooth rows: complex variation with non-dietary variables" goes through and investigates the effect of tooth position in reptiles on dental microwear analysis. The authors found that some taxa had microwear variables that did not vary along the tooth row, while others (n=5) exhibited variation along the tooth row. This study helps to expand our knowledge of how non-dietary variables such as tooth position, bite force, and behavior can affect dental microwear interpretations in reptilians. I recommend minor revisions based upon my comments below. The majority of my comments are minor with perhaps one exception on expanding the discussion to include the taxa that did not exhibit statistical differences along the tooth row.

Comments:

Methods: How do you define the five regions sampled? How did you differentiate these out-levers among different taxa? Looking at figure 1, these out-levers are not the same relative distance between *Varanus rudicollis* and *Alligator mississippiensis*. Are the out-levers the same among all varanids and crocodylians? If so, it would be beneficial to state that clearly in the methods.

It would be helpful to explain the texture variables, including how they are created and what they typically tell us about diet so the reader can be better informed about connecting specific textural variables to diet/food handling/etc. among teeth at different out-levers.

Section 2.6 Statistical Analysis: There is nothing here on how you ran your PCA (I assume it is a PCA, I cannot find anywhere that specifies). How did you run it? I also suggest running a discriminant function analysis instead, since your presumed goal is to categorize the unknown diets along different parts of the tooth row into known dietary categories. You can then use the

Figures: In all figure captions, you need to italicize the scientific names.

Page 12 line 260: "the OL2–5 texture data of species where were the first null hypothesis...." I am not sure what you mean here.

Section 3.2: You go through each taxon individually but basically say the same thing (the first two sentences are basically the same thing word for word for each paragraph). I suggest combining these paragraphs together to reduce the repetitiveness of this section.

Section 3.3 can also be condensed for clarity.

Page 20 line 473: I would change "predate" to another term (I originally read it as time rather than predation). Perhaps change it to "predate upon", which would get rid of the possible confusion.

Page 22, line 523: Please refer to the figure here and elsewhere to point readers to the correct texture-dietary space graph.

Page 23, line 550–552: If you are sampling isolated teeth, how do you propose to identify the original location within the jaw? This is especially difficult with taxa with more homogenous dentitions.

Figures 13–17: You have the letters a–e in each of these figures but no explanation for what these mean. What is the significance of these letters? Please provide a key/legend or in the figure caption.

It is a bit confusing that you only plot five species, even though you sampled more than this (13, I believe). While I understand you are only including species that you find statistical differences among out-levers, it is not expressly stated that you only completed your statistical analysis on these specimens. Due to this, it feels incomplete without any further discussion of these taxa. If you plot the taxa where you did not find statistical difference among the out-levers against your comparative dataset, where would those points fall? How much variation would you see? Just looking at the PCA, there is a whole lot of overlap among dietary guilds in the comparative dataset, so I am wondering if the variation you see among out-levers in your five taxa you plot is similar to the variation you would see if you plot the remainder of the taxa. I would suggest running the PCA (or DFA) for the remaining taxa, and include it (if not in the main body of the text, in the supplemental material). A measure of variation within each out-lever and each taxa would also be useful to help determine the validity of using different tooth loci. I also suggest a discussion on the additional eight taxa that you did not find statistical differences. Why? Are there any commonalities among those taxa that will allow you to predict when you need to take into consideration tooth position or not? If not, why do you think those taxa did not differ?

Appendix B

Reviewer Comments Responses

We thank the reviewers for their reviews and helpful suggestions to improve clarity. We have adopted most of them and a full response is presented below. Reviewer comments and issues are written in **red**, with our responses written in **black** and indented for visual clarity.

“Bestwick et al. Revised_Tracked” shows our updates as tracked changes. “Bestwick et al. Revised_Clean” has the updates incorporated within the document.

Reviewer 1 Comments

This is a long-awaited carefully done look at microwear in squamates and crocodylians. The authors were careful not to overreach in their interpretations, which is otherwise a tendency with microwear. I applaud their care in making hypotheses clear and actually tested with good control over variables not being tested. This opens a field of study that is overdue for work, and is a pattern for the lab leader Purnell.

I cannot find any major revisions needed, and only one minor revision appears worth any effort to make. Near lines 161-162 the authors refer to sampling the non-occlusal labial surfaces as close to the apex as possible. This needs two points of clarification: 1) how is a non-occlusal surface area identified/defined? 2) As close to for some broken teeth (even if broken post-mortem) might be very close to the base. Do you have a criterion for rejection of a tooth if the break is too far from the apex? Other than these two issues, I don't see any need for revisions. I urge the editors to accept this paper, I want to cite it.

In response to comment 1, we now include the sentence in section 2.2. “Wear facets that likely formed from tooth-tooth occlusion from the opening and closing of jaws, characterised by their vertical orientation, elliptical shape and parallel features, were not sampled.”

In response to comment 2, we now include the sentence in section 2.2 “Broken teeth that had more than a quarter of its estimated original height missing were not sampled to increase the likelihood that all sampled tooth apices had experienced regular tooth-food contact”.

Reviewer 2 Comments

The manuscript "Dental microwear texture analysis along reptile tooth rows: complex variation with non-dietary variables" goes through and investigates the effect of tooth position in reptiles on dental microwear analysis. The authors found that some taxa had microwear variables that did not vary along the tooth row, while others (n=5) exhibited variation along the tooth row. This study helps to expand our knowledge of how non-dietary variables such as tooth position,

bite force, and behavior can affect dental microwear interpretations in reptilians. I recommend minor revisions based upon my comments below. The majority of my comments are minor with perhaps one exception on expanding the discussion to include the taxa that did not exhibit statistical differences along the tooth row.

Comments:

Methods: How do you define the five regions sampled? How did you differentiate these outlevers among different taxa? Looking at figure 1, these out-levers are not the same relative distance between *Varanus rudicollis* and *Alligator mississippiensis*. Are the out-levers the same among all varanids and crocodylians? If so, it would be beneficial to state that clearly in the methods.

We have added the following sentences to the first paragraph of section 2.2 to explain this: The underlying assumption of these regions is that in all sampled taxa, the mesially positioned teeth will be subjected to higher bite forces and the distally positioned teeth will be subjected to lower forces (see section 2.4). Given the substantial differences in body size and bite force between the study reptiles, comparisons of the relative forces experienced by teeth can only be done meaningfully within the tooth row of an individual. As only the teeth that are being analysed, and that the extent of the tooth row along the mandible will vary between taxa, the mesial and distal limits for the five loading regions will also vary between taxa. However, as the differences in wear between teeth in different positions in the same individual that are the prime focus, the absolute differences in the positions of the five regions does not affect the final analysis.

It would be helpful to explain the texture variables, including how they are created and what they typically tells us about diet so the reader can be better informed about connecting specific textural variables to diet/food handling/etc. among teeth at different out-levers.

Much of this information is already given in Bestwick et al. (2019). We now explicitly state in section 2.6 and in the legend of Table 1 that more detailed parameter definitions and more information on how parameter values are derived from surface textures can be found in the supplementary information of Bestwick et al. (2019). This saves needless repetition of previously published figures, tables and text.

In section 3.1, we provide qualitative descriptions of how texture parameters differ between out-lever positions based on the parameters that exhibit significant differences. We do not relate them explicitly to diet at this point as it is unclear if that is a determinant behind the texture differences.

Bestwick J. et al. (2019) *Sci. Rep.* **9**, 11691.

Section 2.6 Statistical Analysis: There is nothing here on how you ran your PCA (I assume it is a PCA, I cannot find anywhere that specifies). How did you run it? I also suggest running a discriminant function analysis instead, since your presumed goal is to categorize the unknown diets along different parts of the tooth row into known dietary categories. **You can then use the**

We now specify in section 2.6 that the reptile texture-dietary space is derived from a principal component analysis. We now explicitly state that this multivariate space was constructed using four ISO parameters that exhibited significant differences between extant reptile dietary guilds. This is sufficient detail to address the reviewer's concerns without producing needless repetition from Bestwick et al. (2019).

Although the final sentence of the comment is truncated, we understand the point that the reviewer is making about the differences between PCAs and DFAs. We considered DFA (or linear discriminant analysis) and its potential use in the context of dietary and tooth row analyses. Indeed, one of the authors has previous publications on microwear texture analysis that employ both PCA and DFA (e.g. Purnell *et al.* 2013; Purnell and Darras 2016). However, this experience led us to prefer PCA, and this approach is consistent with other dietary studies that include extant and/or extinct taxa (Delezenne et al. 2013; 2016; Gill et al. 2014; Purnell et al. 2012; 2013; 2017; Purnell & Darras 2016; Bestwick et al. 2019; Winkler et al. 2019). This preference stems from the fundamental difference between PCA and DFA: PCA requires no a priori assumptions about out-lever position, whereas DFA does. This gives PCA an advantage over DFA for two reasons. First, the initial stage of our analysis was to determine whether microwear textures differ between teeth from different positions of the tooth row. PCA, because it does not assume that teeth assigned to similar out-levers should have similar microwear, does not impose a structure on the multivariate space that reflects the initial assumptions to the degree that DFA does. It allows for the possibility that teeth from the same out-levers do not have similar microwear, rather than 'trying' to allocate them to the same region of multivariate space. If the results of this phase of analysis reveal that there are significant relationships between tooth position and the principal axes that capture the highest proportion of the variation in texture, this can serve as a multivariate framework within which to test hypotheses about out-levers. Second, in the context of long extinct animals for which choices of precise dietary analogues among extant animals can be complex, this framework approach is preferable to one that attempts to classify (such as DFA). There is no need to make any assumptions that the dietary classes of the extant animals capture the same dietary classes to which the extinct taxa belonged. Similarly we do not have to assume that the relationship between dietary class and microwear texture is precisely the same in distantly related extant and extinct taxa. It is important to reiterate here that in the way our study employs PCA the OLs 2-5 do not have any role in structuring the PCA space upon which their position is based. They are not constrained to plot according to

any groupings reflected in the distribution of the OL 1 data, and in this respect our approach takes advantage of one of the most powerful aspects of PCA.

As we note above, a number of studies published in high quality peer reviewed journals, some of which are highly cited, employ the same PCA-based analytical approach that we have used here. While we agree with the reviewer that for some kinds of analysis DFA could be considered to represent a gold standard, we are of the opinion that PCA is more appropriate for our analysis, given the nature of our questions, uncertainties, and the data

Bestwick J. *et al.* (2019) *Sci. Rep.* **9**, 11691.

Delezene L. K. *et al.* (2013) *J. Hum. Evol.* **65**, 282–293.

Gill P. G. *et al.* (2014) *Nature* **512**, 303–305.

Purnell M. A. and Darras L. P. G. (2016) *Surf. Topog.: Metrol. Prop.* **4**, 014006.

Purnell M. A. *et al.* (2012) *J. R. Soc. Interface* **9**, 2225–2233.

Purnell M. A. *et al.* (2013) *J. Zool.* **291**, 249–257.

Purnell M. A. *et al.* (2017) *Biosurf. Biotribol.* **3**, 184–195.

Winkler D. E. *et al.* (2019) *Proc. R. Soc. B.* **286**, 20190544.

Figures: In all figure captions, you need to italicize the scientific names.

This has been done for all figure and table legends.

Page 12 line 260: "the OL2–5 texture data of species where were the first null hypothesis...." I am not sure what you mean here.

We thank the reviewer for spotting this error. The sentence has been corrected for clarity.

Section 3.2: You go through each taxon individually but basically say the same thing (the first two sentences are basically the same thing word for word for each paragraph). I suggest combining these paragraphs together to reduce the repetitiveness of this section.

This section has been condensed by over 120 words to reduce repetitiveness.

Section 3.3 can also be condensed for clarity.

This section has been condensed slightly where possible, but we feel in the interest of transparency that the results of the ANOVA and pairwise testing are best reported in the text. This inevitably does create some repetitiveness in the text but this does not inhibit the reader's ability to interpret the results of the statistical tests.

Page 20 line 473: I would change "predate" to another term (I originally read it as time rather than predation). Perhaps change it to "predate upon", which would get rid of the

possible confusion.

Done. Sentence now reads “For example, *V. rudicollis* predate upon taxa much smaller than themselves...”.

Page 22, line 523: Please refer to the figure here and elsewhere to point readers to the correct texture-dietary space graph.

This has been done throughout section 4.2 for all mentioned study reptiles and in all cases the correct figure has been referenced.

Page 23, line 550–552: If you are sampling isolated teeth, how do you propose to identify the original location within the jaw? This is especially difficult with taxa with more homogenous dentitions.

No change. We believe it is beyond the scope of our study to provide instructions on identifying the original *in situ* locations of isolated reptile teeth. As our main study aim is to determine whether standardised sampling positions are needed for reptile DMTA, we therefore provide the recommendation that such sampling procedures are best adopted.

Figures 13–17: You have the letters a–e in each of these figures but no explanation for what these mean. What is the significance of these letters? Please provide a key/legend or in the figure caption.

Letters a–e in the texture-dietary spaces correspond to the presented tooth surface textures in previous figures to show where they plot in the texture-dietary spaces. E.g. the *Alligator mississippiensis* datum points highlighted in figure 13 correspond to the textures shown in figure 2. The legends for figures 13–17 have been updated to explicitly state this to minimise confusion.

It is a bit confusing that you only plot five species, even though you sampled more than this (13, I believe). While I understand you are only including species that you find statistical differences among out-levers, it is not expressly stated that you only completed your statistical analysis on these specimens. Due to this, it feels incomplete without any further discussion of these taxa. If you plot the taxa where you did not find statistical difference among the out-levers against your comparative dataset, where would those points fall? How much variation would you see? Just looking at the PCA, there is a whole lot of overlap among dietary guilds in the comparative dataset, so I am wondering if the variation you see among out-levers in your five taxa you plot is similar to the variation you would see if you plot the remainder of the taxa. I would suggest running the PCA (or DFA) for the remaining taxa, and include it (if not in the main body of the text, in the supplemental material). A

measure of variation within each out-lever and each taxa would also be useful to help determine the validity of using different tooth loci. I also suggest a discussion on the additional eight taxa that you did not find statistical differences. Why? Are there any commonalities among those taxa that will allow you to predict when you need to take into consideration tooth position or not? If not, why do you think those taxa did not differ?

We are now more explicit about plotting data from only the five species where our first null hypothesis could be rejected (microwear textures do not differ between teeth from different positions of the tooth row), into the texture-dietary space. This has been done in the last sentence of the introduction. We also mention this in paragraph 2 of section 2.6 which, together, minimises confusion for readers. We only plotted these five species due to the experimental framework we set out for the manuscript, i.e texture differences along the tooth row indicates possible dietary partitioning along the jaws and thus warrants further investigation. We therefore had little scientific justification to do this for the eight species that did not show texture differences along tooth rows. This would have resulted in a huge additional body of work that would have unlikely yielded any meaningful trends. However, in the interest of transparency and reproducibility we have provided the PC1 and PC2 data of every sampled tooth from all 13 reptile species within Table S1 so that readers can further explore the data should they wish.

In the first paragraph of section 4.1, we state that there are several commonalities between the reptiles and do and do not exhibit texture differences along tooth rows e.g. dietary guild, degree of tooth heterodonty and behavioural usage of the jaws. This is why in our conclusions we recommend that standardised sampling positions for taxa with non-occlusal dentitions are best adopted. We refrained from discussing why no discernible tooth texture differences were found in eight of our reptile species because the endogenous and exogenous mechanisms that underlie tooth texture formation remain poorly understood. We therefore feel that postulating reasons for the absence of texture differences would amount to little more than conjecture and thus obscure the structure and message of our narrative. At the end of the first paragraph of section 4.1, we now provide several recommendations for future investigations into these taxa, such as larger sample sizes. This better highlights that we are not yet at a position to robustly explain why no texture differences were found along the tooth rows of these taxa and provides a clearer path for what can be done to rectify the issue.